# Repurposing an endogenous degradation domain for antibody-mediated disposal of cell-surface proteins

Janika Schmitt[1,4,5], Emma Poole[1,2], Ian Groves [1,6], David J Owen [3✉], Stephen C Graham [2✉], John Sinclair [1✉] & Bernard T Kelly [3✉]

## Abstract

**The exquisite specificity of antibodies can be harnessed to effect targeted degradation of membrane proteins. Here, we demonstrate targeted protein removal utilising a protein degradation domain derived from the endogenous human protein Proprotein Convertase Subtilisin/Kexin type 9 (PCSK9). Recombinant antibodies genetically fused to this domain drive the degradation of membrane proteins that undergo constitutive internalisation and recycling, including the transferrin receptor and the human cytomegalovirus latency-associated protein US28. We term this approach PACTAC (PCSK9-Antibody Clearance-Targeting Chimeras).**

**Keywords** Human Cytomegalovirus; PACTAC; PCSK9; Targeted Protein Degradation
**Subject Categories** Membranes & Trafficking; Methods & Resources; Post-translational Modifications & Proteolysis

## Introduction

Antibodies are well-known for their exquisite target specificity, and are used as therapeutics to treat a wide variety of human diseases. Because of their large size ( ~ 150 kDa) and inability to cross the cell's plasma membrane (PM), antibodies are generally not able to engage cytoplasmic targets in vivo, being instead well-suited to targeting secreted proteins such as the proinflammatory cytokine TNFα or transmembrane proteins such as the Erbb2 receptor tyrosine kinase / Her2 (Chames et al, 2009). Typically, antibody therapeutics work by blockade of a functional binding surface; by recruitment of immune effector functions leading to target cell killing (ADCC, antibody-dependent cellular cytotoxicity); or by mediating endolysosomal delivery of cytotoxic drug payloads to kill cancerous or infected cells (ADCs, antibody-drug conjugates) (Drago et al, 2021). Nonetheless, many potential transmembrane protein targets, such as G-Protein Coupled Receptors (GPCRs) (Jo

and Jung, 2016) or voltage-gated sodium channels (Wulff et al, 2019), have proven difficult to drug effectively with antibodies, and some targets of ADCs evade delivery to endolysosomes by efficient recycling to the PM (Hammood et al, 2021).

To overcome these difficulties, recent advances have used antibodies to drive target degradation, inspired by the concept of Proteolysis Targeting Chimeras (PROTACs)(Békés et al, 2022). PROTACs (whether small molecules or proteins (Clift et al, 2018)) forcibly recruit cytoplasmic E3 ubiquitin ligases to a target to stimulate its ubiquitination and subsequent proteasomal degradation, and thus have emerged as a key approach to target 'undruggable' cytoplasmic proteins. Polyubiquitination of the cytosolic domains of transmembrane proteins marks them for degradation in endolysosomes rather than proteasomes, a process mediated by ESCRTs (Endosomal Sorting Complexes Required for Transport). AbTACS/PROTABs (Marei et al, 2022; Cotton et al, 2021) take advantage of this mechanism by forcibly recruiting transmembrane E3 ubiquitin ligases to target receptors with bispecific antibodies, resulting in ubiquitination of the target's cytosolic domains and subsequent endolysosomal degradation. In contrast, LYTACs (Banik et al, 2020) hijack an endogenous mechanism for delivery of lysosomal hydrolases by labelling antibodies with sugars that are recognized by the cation-independent mannose-6-phosphate receptor (CIMPR), leading to co-trafficking of the antibody and its bound target together with CIMPR to endolysosomes, where the target is degraded.

To expand the antibody-mediated protein degradation toolbox, we wished to find a protein tag that could be fused genetically to antibodies or antibody fragments to direct targets to endolysosomal degradation without the need for bispecific antibody formatting or chemical modification. We took as our starting point an endogenous mechanism for transmembrane protein degradation mediated by the secreted human protein PCSK9 that is apparently distinct from other known endolysosomal delivery pathways. PCSK9 suppresses low-density lipoprotein (LDL) receptor (LDLR), which is responsible for LDL-cholesterol homeostasis, by a poorly understood mechanism that reroutes LDLR to endolysosomes for degradation (Fig. 1A) (Zhang et al, 2007) independent of ESCRT and ubiquitination (Wang et al, 2012). The N-terminal domain of PCSK9 binds tightly to LDLR at the PM and PCSK9 lacking the

---

[1]Department of Medicine, Cambridge Institute of Therapeutic Immunology and Infectious Disease, University of Cambridge, Hills Road, CB2 0SP Cambridge, UK. [2]Department of Pathology, University of Cambridge, Tennis Court Road, Cambridge CB2 1QP, UK. [3]Cambridge Institute for Medical Research, Keith Peters Building, Hills Road, Cambridge CB2 0XY, UK. [4]Present address: Faculty of Medicine, Charité Berlin, 10117 Berlin, Germany. [5]Present address: Faculty of Medicine, University of Heidelberg, 69210 Heidelberg, Germany. [6]Present address: Infection Biology, Global Center for Pathogen and Human Health Research, Lerner Research Institute, Cleveland Clinic, Cleveland, OH 44195, USA. ✉E-mail: djo30@cam.ac.uk; scg34@cam.ac.uk; js152@cam.ac.uk; btk1000@cam.ac.uk

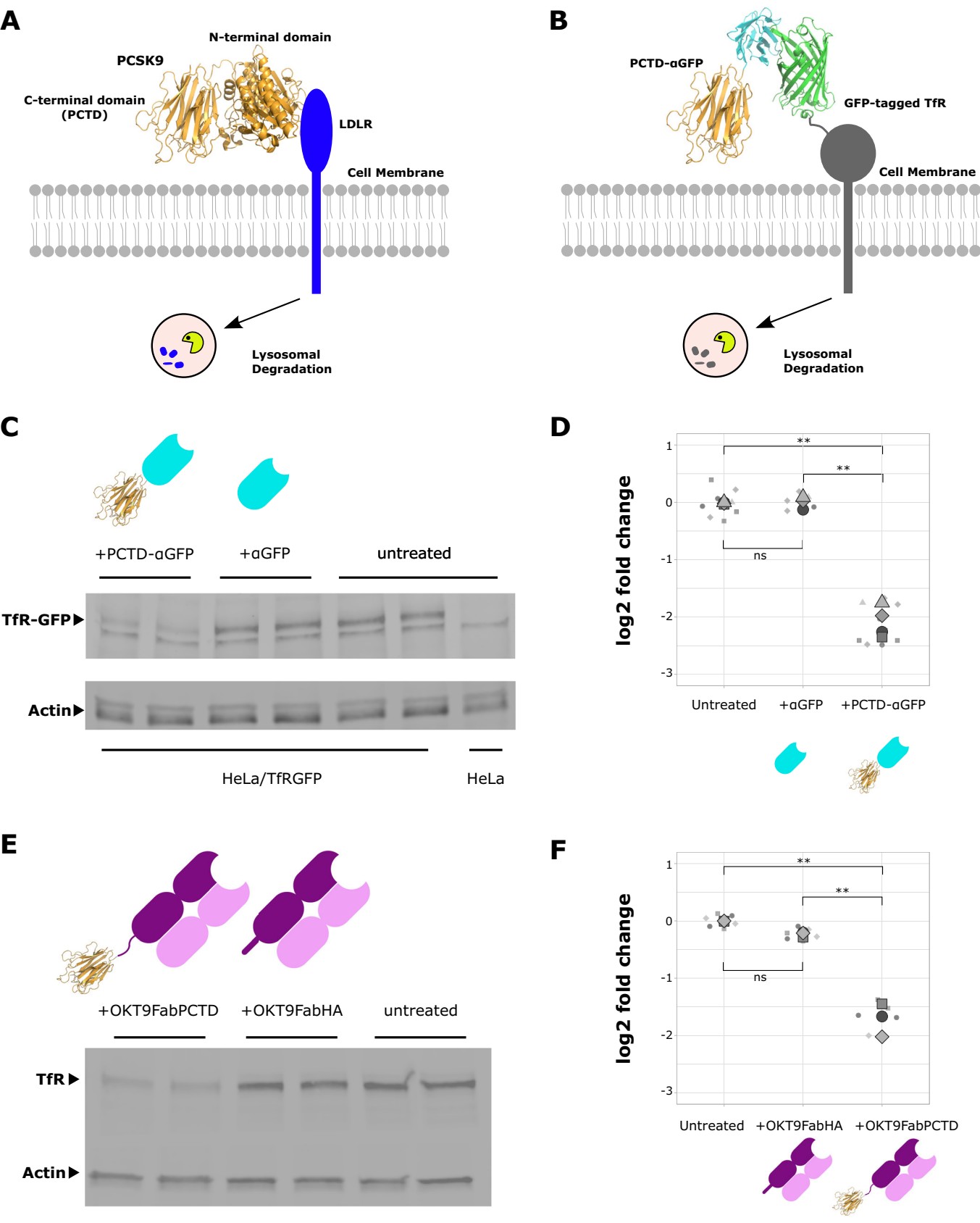

**Figure 1.  Targeted degradation by PCTD.**

(A) Degradation of LDLR by endogenous PCSK9. (B) PCTD-mediated degradation of target transmembrane protein. (C) PCTD-αGFP, but not αGFP alone, drives degradation of stably expressed TfR-GFP in HeLa cells (assessed by GFP immunoblot). PCTD-αGFP and αGFP are depicted as cartoons with αGFP in cyan and PCTD in gold. (D) TfR-GFP degradation assessed by densitometry. Data shown as Superplot (Goedhart, 2021): biological replicates differ by shape; large symbols indicate means. Differences in means were assessed by one-way ANOVA with Tukey's method for multiple comparisons: ns, $p \geq 0.05$; **$p < 0.01$. (E) OKT9FabPCTD, but not OKT9FabHA, drives degradation of endogenous TfR in HeLa cells. (F) TfR degradation assessed by densitometry. Data shown as Superplot (as in (D)). OKT9Fab constructs are depicted with cartoons; heavy chains are shown in dark purple, light chains in light purple with PCTD in gold. Differences in means were assessed by one-way ANOVA with Tukey's method for multiple comparisons: ns, $p \geq 0.05$; **$p < 0.01$. Source data are available online for this figure.

Cys/His-rich C-terminal domain is unable to stimulate LDLR degradation (Zhang et al, 2008). Here, we show that the ~25 kDa PCSK9 C-terminal domain, which we term PCTD, is sufficient to drive lysosomal degradation of a target when fused to antibody fragments that mediate target recognition (Fig. 1B), an approach we term PACTAC (PCSK9-Antibody Clearance-Targeting Chimeras). We show that targets suitable for PACTAC are transmembrane proteins that internalise and recycle through endosomal compartments, and demonstrate targeted degradation of two therapeutic antibody targets, Transferrin Receptor 1 (TfR) and the human cytomegalovirus (HCMV) latency-associated protein US28.

## Results

### Antibody fragments fused to PCTD drive degradation of tagged and endogenous transferrin receptors

We began by creating a HeLa cell line stably expressing a reporter construct, Transferrin Receptor 1 (TfR) tagged on the extracellular face with GFP (Fig. 1B). Notably, TfR is overexpressed in proliferating cells in order to satisfy increased demand for iron (Candelaria et al, 2021). Overexpression of TfR is seen in human cancers (Sutherland et al, 1981), including most breast cancers (Singh et al, 2011) and various haematological malignancies (Guo et al, 2021). Thus, TfR overexpression has been exploited with a range of therapeutic approaches. For instance, an anti-TfR antibody has been used for tumour-directed delivery of immunotoxins (Wang et al, 2022), and TfR-directed chimeric antigen receptor (CAR) T-cells are effective in mouse models of T-cell acute lymphoblastic leukemia (Guo et al, 2021). To determine if PCTD could drive degradation of the reporter TfR construct, we purified from HEK-293F cells a chimeric protein composed of an anti-GFP nanobody (Rothbauer et al, 2008) (αGFP) fused to PCTD, termed PCTD-αGFP, and, as a control, we purified αGFP from *E. coli*. The TfR-GFP reporter cell line was treated with PCTD-αGFP, αGFP, or left untreated. After overnight incubation, the cells were lysed and cell extracts probed by immunoblot for the presence of TfR-GFP. PCTD-αGFP drove a large (4.6-fold; 78%) and significant ($p < 0.01$) reduction of TfR-GFP, while no significant degradation was stimulated by αGFP alone (Fig. 1C,D).

To demonstrate the targeted degradation of endogenous TfR by PACTAC, the anti-TfR antibody OKT9 (Sutherland et al, 1981) was fused to PCTD. This was achieved by expressing and purifying a Fab fragment of OKT9 with PCTD fused (via a 35-residue glycine-serine linker) to the C terminus of the truncated heavy chain (OKT9FabPCTD) via co-transfection of HEK-293F cells with vectors encoding the light chain (VL-CL) and the heavy chain-PCTD fusion (VH-CH1-PCTD). As a control, a modified version of

this construct with the PCTD domain replaced by a dual HA (hemagglutinin) tag, termed OKT9FabHA, was expressed and purified. HeLa cells were treated overnight with OKT9FabPCTD, OKT9FabHA, or left untreated before lysis and analysis by immunoblot (Fig. 1E,F). OKT9FabPCTD drove a significant ($p < 0.01$) and substantial reduction (3.3-fold; 69%) of TfR compared to untreated cells, similar to the degradation of PD-L1 observed with AbTACs (Cotton et al, 2021). OKT9FabHA caused a modest decrease (1.2-fold, 15%) that was not significantly different from untreated cells. These results confirm that PCTD can drive the degradation of proteins expressed at endogenous levels, and that degradation is effective when PCTD is conjugated to either nanobodies or antibody Fab fragments.

### Characterisation of PCTD-mediated degradation

To extend our investigation of PCTD-mediated TfR degradation, we first verified that the mechanism is not exclusive to HeLa cells by demonstrating OKT9FabPCTD-mediated TfR degradation in a different cell type, the breast cancer SK-BR-3 cell line. We performed a dose-response experiment by titrating the amount of OKT9FabPCTD used to treat SK-BR-3 cells overnight and assessing TfR degradation by Western Blot (Fig. 2A), which demonstrated substantial TfR degradation in this cell type at an OKT9FabPCTD concentration of 10 nM and almost complete TfR loss at an OKT9FabPCTD concentration of 100 nM. Next, we performed a degradation time course experiment in the SK-BR-3 cell line, treating cells with 25 nM OKT9FabPCTD (Fig. 2B), which caused progressive loss of TfR that could be fitted to an exponential decay curve with a characteristic degradation rate of 22% per hour (Fig. 2B) with a plateau of ~83% degradation. In contrast, treatment of SK-BR-3 cells for 24 h with 25 nM OKT9FabHA results in comparatively little loss of TfR (~20%) (Fig. 2C).

The efficiency of targeted degradation by PROTABs (in which a bispecific antibody recruits a transmembrane E3 ligase to a target receptor) has been shown to be sensitive to antibody formatting; for instance, superior degradation was observed for a C-terminal fusion of an anti-target scFv to the heavy chain of a whole anti-ligase IgG, compared to a traditional 'knobs-into-holes' bispecific IgG format (Gramespacher et al, 2022). Similarly, we asked if the position of the PCTD domain fusion in our OKT9-PTCD construct affects the efficiency of TfR degradation. The OKT9FabPCTD construct contains a PCTD domain fused to the C-terminus of the heavy chain $C_H1$ domain and separated from the heavy chain by a 35-residue glycine-serine linker. We constructed a panel of three further constructs in which PCTD was fused by an identical linker to the C-terminus of the light chain $C_L$ domain, or without linkers to the C-termini of the heavy chain $C_H1$ or $C_L$ domains. We expressed and purified these antibody fragments, treated SK-BR-3

**A**

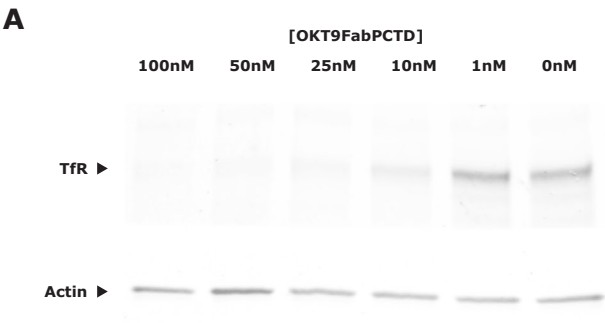

**B**

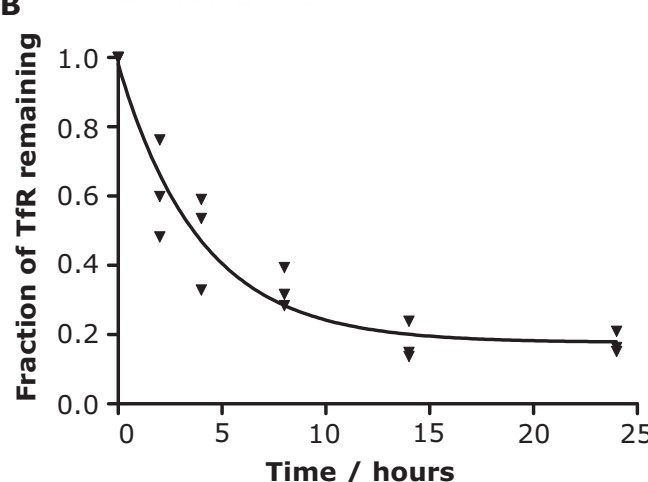

**C**

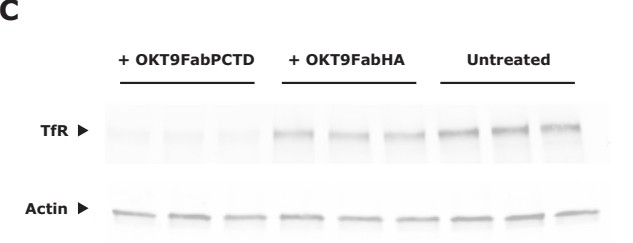

**Figure 2. Characterization of PCTD-mediated degradation in SK-BR-3 cells.**

(A) Dose-response experiment. SK-BR-3 cells were treated with OKT9FabPCTD at the concentrations shown and incubated overnight before TfR degradation was assessed by Western Blot. (B) Time course of TfR degradation in SK-BR-3 cells. Cells were treated with 25 nM OKT9FabPCTD and incubated for various lengths of time. TfR degradation was assessed by Western Blot and densitometry and plotted as fractional TfR remaining compared to mean of untreated controls; three biological replicates are shown for each time point. The degradation curve was fitted to the equation $y = D_{max}e^{(-\lambda.t)} + m$, where y is the fractional TfR remaining, $D_{max}$ is the maximal fractional TfR loss, t is incubation time in hours, m is the plateau or minimal remaining fraction of TfR, and $\lambda$ is the degradation rate. The fitted parameters were $D_{max}$ (0.80), $\lambda$ (0.25 $h^{-1}$) and m (0.17); goodness of fit: $R^2 = 0.93$. (C) SK-BR-3 cells were treated for 24 h with 25 nM OKT9FabPCTD or OKT9FabHA or left untreated as indicated (3 biological replicates per treatment), and TfR degradation assessed by Western Blot as in (B). Source data are available online for this figure.

cells for 4 h with 25 nM of each construct and assessed TfR degradation (Fig. EV1). This experiment demonstrated no significant differences in TfR degradative efficiency between the different OKT9 antibody formats tested. However, it remains a

possibility that, for other target epitopes, antibody-PCTD format might affect the efficiency of target degradation.

## PCTD-mediated endolysosomal degradation requires internalisation and endosomal trafficking

Many transmembrane proteins undergo clathrin-mediated endocytosis (i) (Fig. 3A) and recycling through a complex variety of endosomal compartments back to the PM (ii) (Fig. 3A) (Maxfield and McGraw, 2004) or the *trans*-Golgi network (TGN) (iii) (Fig. 3A). Such traffic is typically mediated by unstructured short linear motifs in the cytoplasmic domains of transmembrane proteins (Traub and Bonifacino, 2013). TfR, for instance, is efficiently internalised from the cell surface via a YTRF motif but constitutively recycles back to the PM (Harding et al, 1983). LDLR, the endogenous target of PCSK9, undergoes constitutive internalisation and recycling via an NPxY motif (Bartuzi et al, 2016).

To probe whether PCTD-driven degradation requires target internalisation and recycling, cell lines were generated expressing a chimeric reporter comprising an extracellular CD8 α-chain (chosen as a 'neutral', folded domain) tagged with the β-catenin-derived BC2 peptide tag (Braun et al, 2016a), a single transmembrane helix, a *variable* cytoplasmic domain, and an intracellular EGFP for fluorescence detection (Fig. 3B). This design allowed all cell lines to be treated with the same reagents, eliminating a potential source of variation. PCTD fused to an anti-BC2 nanobody (Braun et al, 2016a), termed PCTD-αBC2, was purified from HEK-293F cells and αBC2 was purified from *E. coli*. The variable cytoplasmic domains were as follows: *NPxY* contains an NPxY-type motif that drives both internalisation by clathrin-mediated endocytosis (Chen et al, 1990) and PM recycling (Steinberg et al, 2012); *ΔMOTIF* lacks known internalisation motifs; and *TGN* contains a YQRL motif known to direct efficient internalisation from the PM and rapid rerouting from early endosomal compartments to the TGN (Bos et al, 1993). All three constructs were accessible by extracellular ligands, confirmed by uptake of fluorescent anti-CD8 antibody (Fig. EV2).

Cell lines were treated overnight with PCTD-αBC2, αBC2 alone, or left untreated, and cellular GFP fluorescence was analysed by flow cytometry to measure degradation of the reporter relative to untreated cells (Fig. 3C). Treatment with αBC2 alone did not cause significant degradation of any of the reporters (Fig. 3D). Treatment with PCTD-αBC2 provoked significant degradation of the *NPxY* reporter (1.5-fold, 34%), but not of the *ΔMOTIF* reporter (Fig. 3D). The *TGN* reporter displayed an intermediate phenotype, being significantly but modestly degraded by PCTD-αBC2 (1.1-fold, 9%). These data suggest that PCTD degrades targets that recycle through 'later' endosomal compartments more effectively than those that traffic through the PM before retrieval to the TGN (Hirst et al, 2012). More generally, trafficking itinerary might influence PCTD-mediated degradation efficiency; the reported affinities of OKT9 (Ferrero et al, 2021) and αBC2 (Braun et al, 2016b) for their targets are similar (4 nM vs 1.4 nM respectively) whereas the extent of degradation observed differs (69% vs 34% in HeLa cells, respectively).

To assess whether PCTD drives lysosomal destruction of targets, degradation experiments were performed in the presence or absence of bafilomycin A1, a compound that inhibits protein hydrolysis by blocking lysosomal acidification. After pre-incubation of *NPxY* reporter cells with carrier (DMSO) (Fig. 4A) or

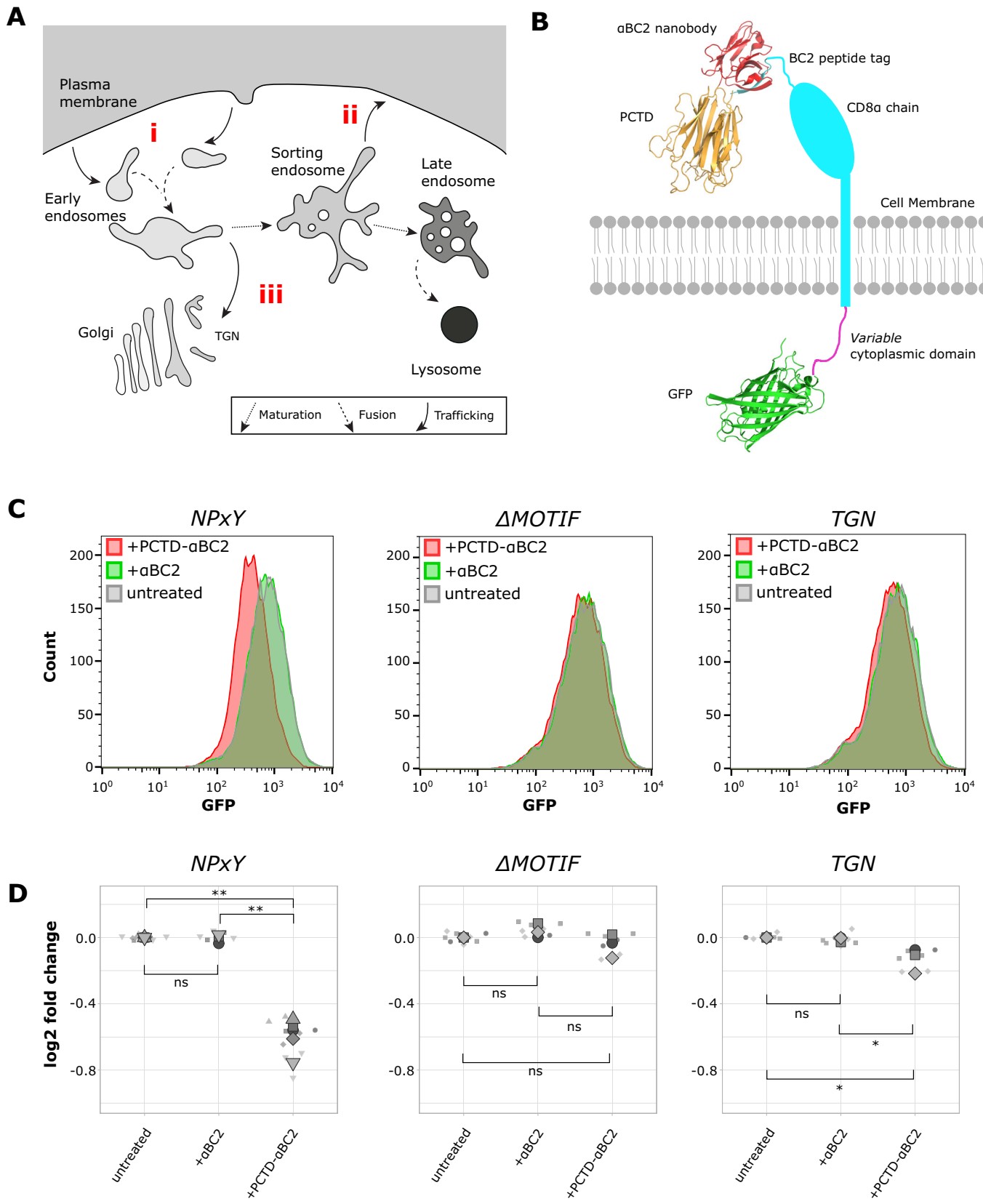

**Figure 3.   Dependence of PCTD-mediated degradation on passage through the endosomal system.**

(A) Simplified schematic of endosomal trafficking. (B) Schematic of reporter constructs. (C) Flow cytometric analysis of PCTD-mediated degradation. Representative traces from at least three biological replicates. (D) Log$_2$-fold change in mean fluorescence with treatment compared to mean fluorescence of untreated is depicted for all experiments as a Superplot (Goedhart, 2021). Differences in means were assessed by one-way ANOVA with Tukey's method for multiple comparisons: ns, $p \geq 0.05$; *$p < 0.05$; **$p < 0.01$. Source data are available online for this figure.

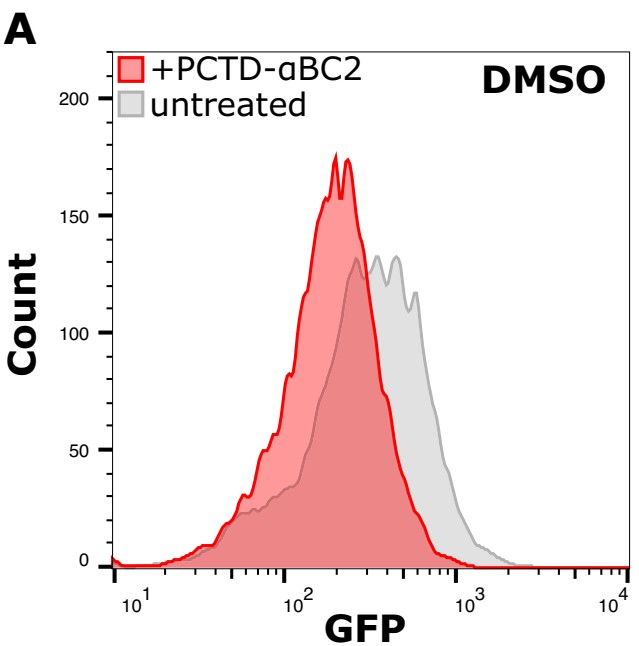

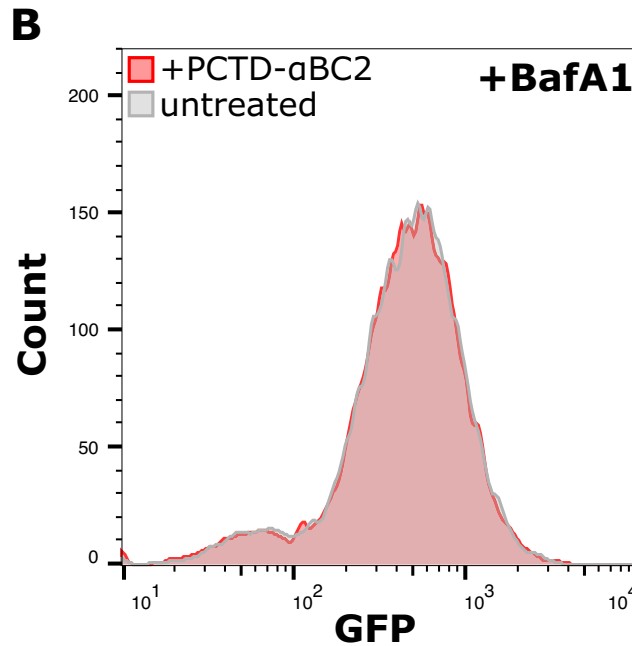

**Figure 4.   Dependence of PCTD-mediated degradation on lysosomal acidification.**

(A, B) *NPxY* reporter cells were pre-incubated with DMSO (A) or Bafilomycin A1 (B) for 2 h, treated overnight with PCTD-αBC2 (red) or PBS (grey), and cellular GFP fluorescence analysed by flow cytometry. Data representative of two biological replicates.

bafilomycin A1 (Fig. 4B), the cells were treated overnight by supplementing the medium with PCTD-αBC2 or left untreated. PCTD-αBC2 was unable to drive the degradation of reporter in the presence of bafilomycin A1 (Fig. 4B), confirming that PCTD stimulates the lysosomal degradation of targets.

## PCTD directs the degradation of HCMV latency-associated protein US28

Next, we sought to apply PACTAC technology to a different medically relevant target, the human cytomegalovirus (HCMV) protein US28. HCMV is a beta-herpesvirus with an estimated global seroprevalence of 60 to 90% (Griffiths and Reeves, 2021). While primary infection with HCMV is often subclinical, it can cause severe end-organ disease in immunocompromised individuals (Griffiths and Reeves, 2021). A hallmark of HCMV infection is the establishment of a lifelong latent infection in undifferentiated cells of the myeloid lineage (Sinclair and Sissons, 2006; Poole and Sinclair, 2015), which cannot be targeted with current HCMV antivirals (Krishna et al, 2019; Perera et al, 2021). Latency is characterized by the suppression of viral immediate early (IE) gene expression and the absence of infectious virion production (Poole and Sinclair, 2015). Reactivation from latency, which

depends on the activation of viral IE gene expression, results in the production of infectious particles. The initial switch from latency to lytic infection is controlled by a single viral promoter, the major immediate early promoter (MIEP), which itself is controlled by epigenetic repression (Elder and Sinclair, 2019). US28, a viral chemokine receptor expressed during latent and lytic infection (Krishna et al, 2018), is crucial for mediating the repression of the MIEP during latency. We recently developed a virus-specific nanobody (Vun100bv) that functions as a partial inverse agonist of US28 (De Groof et al, 2021) and induces the reactivation of HCMV from latency. The induction of viral IE gene expression offers therapeutic potential, as reactivation allows infected cells to be recognized by HCMV-specific cytotoxic T-cells ("shock and kill") (De Groof et al, 2021; Wills et al, 2015).

US28 undergoes rapid constitutive endocytosis and recycling back to the PM (Fraile-Ramos et al, 2001), suggesting that it might be a suitable target for PCTD-driven degradation. Vun100bv was fused to the C terminus of PCTD to generate a virus-specific nanobody conjugate termed PCTD-Vun100bv. First, we assessed the ability of PCTD-Vun100bv to target and degrade US28 in fibroblasts, a cell type that permits HCMV lytic infection. Fibroblasts were infected with HCMV encoding GFP-tagged US28 (US28-GFP-HCMV) and subsequently treated with PCTD-

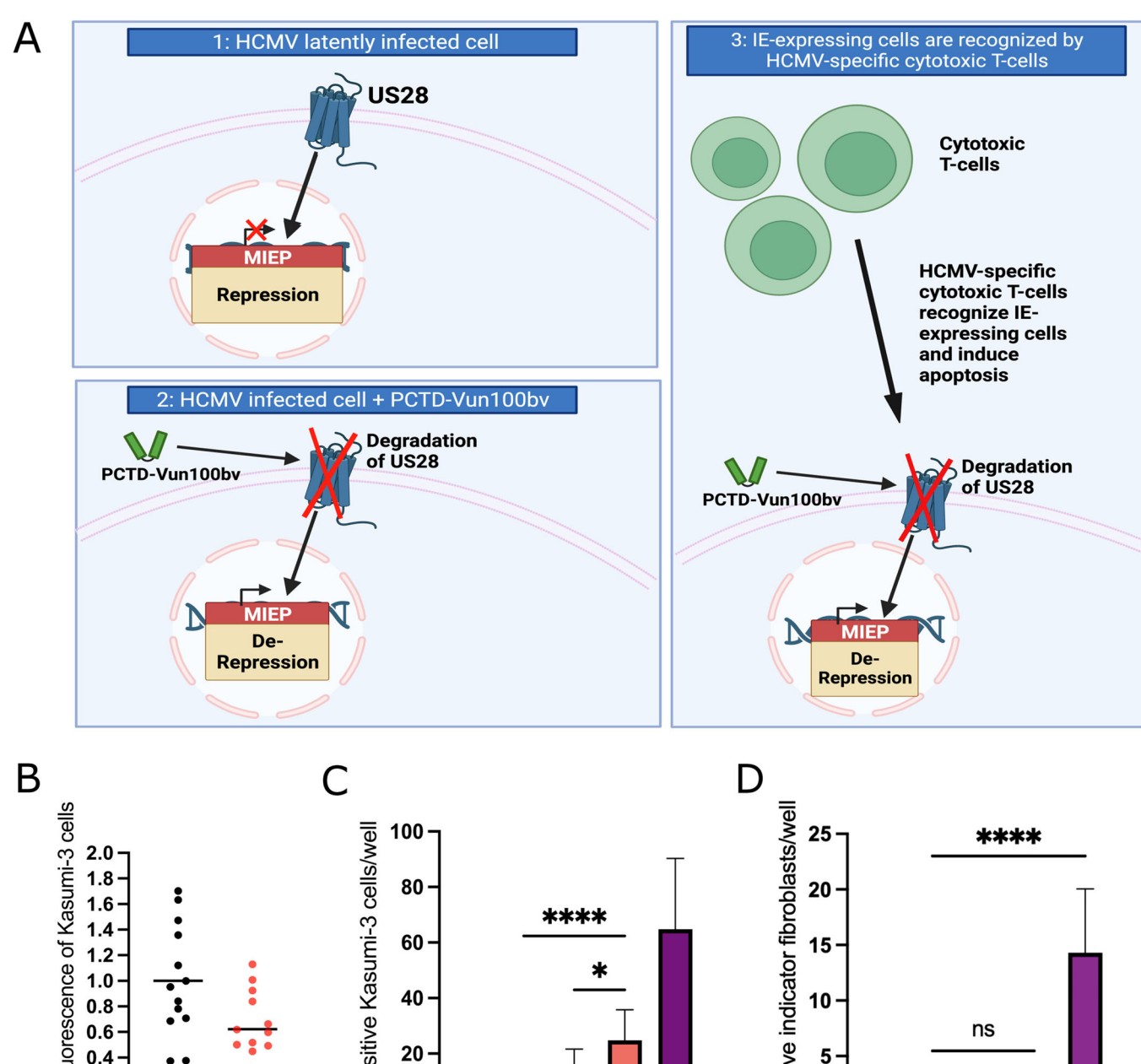

Vun100bv, the non-US28-targeting PCTD-αBC2 nanobody, or left untreated. US28 expression levels were monitored using live-cell fluorescence microscopy. As early as 60 min after treatment with PCTD-Vun100bv, a decrease in the level of US28-specific fluorescence can be observed compared with untreated or PCTD-αBC2 treated cells (Fig. EV3). To confirm that PCTD was responsible for this effect, we directly compared the effect of

PCTD-Vun100bv and Vun100bv on the level of US28 expression. Fibroblasts were infected with US28-GFP-HCMV and then treated with PCTD-Vun100bv, Vun100bv, or left untreated. We observed a distinct loss of total GFP levels in PCTD-Vun100bv treated cells, whereas treatment with Vun100bv had little effect on US28, suggesting that the PCTD modification is responsible for receptor degradation (Fig. EV4).

**Figure 5. PCTD-Vun100bv induces US28 degradation in Kasumi-3 cells and prevents the establishment of latency without driving full virus reactivation.**

(A) Schematic depicting the potential use of PCTD-Vun100bv as a "shock-and-kill" therapeutic. Created with BioRender.com. (B) Kasumi-3 cells were infected with recombinant HCMV-US28-GFP and were left untreated or were treated with PCTD-Vun100bv two days post-infection and imaged three days post-treatment. The reporter fluorescence (assessed by fluorescence microscopy) was normalized to the average reporter fluorescence of untreated cells. (C, D) Kasumi-3 cells were infected with HCMV-IE2-eYFP and, two hours post-infection, were treated with Vun100bv, PCTD-Vun100bv, or PMA. (C) IE-positive nuclei were counted three to four days post-treatment. (D) Four days post-treatment, the supernatant was transferred onto fibroblasts. IE-positive cells were counted four days after supernatant transfer. Data information: Statistical analyses were performed using one-way ANOVA followed by Tukey's multiple comparison test (C) or Kruskal-Wallis-Test followed by Dunn's multiple comparison test (D): ns, $p \geq 0.05$; *$p < 0.05$; ****$p < 0.0001$. Data show individual data points and mean representative of two biological replicates (B) or mean and SD of four (C) and three (D) biological replicates. Source data are available online for this figure.

Since fibroblasts support only lytic infection, we next studied PCTD-Vun100bv treatment in cell types supporting HCMV latency. US28 signalling is essential to repress the HCMV MIEP to establish and maintain latency in Kasumi-3 cells (Fig. 5A.1) (Humby and O'Connor, 2015a). We hypothesized that like the unmodified nanobody, Vun100bv, PCTD-Vun100bv treatment would drive IE gene expression (Fig. 5A.2), which in turn would plausibly allow for the subsequent killing of infected cells by HCMV-specific cytotoxic T-cells (Fig. 5A.3). Thus, we repeated our analyses in the Kasumi-3 cell line, a CD34+ myeloblastic cell line known to support HCMV latency and reactivation (O'Connor and Murphy, 2012; Albright and Kalejta, 2013). PCTD-Vun100bv treatment promoted degradation of US28, indicated by the reduction of fluorescence intensity (Fig. 5B). To determine if US28 degradation by PCTD-Vun100bv prevents the establishment of latency, we infected Kasumi-3 cells with an HCMV-IE2-eYFP virus that allows the detection of cells expressing lytic IE gene products, and treated cells with nanobodies 2 h post-infection. As a positive control, we treated Kasumi-3 cells with phorbol myristate acetate (PMA), which makes them fully permissive for lytic HCMV infection (O'Connor and Murphy, 2012). PCTD-Vun100bv induced IE expression levels ~3-fold higher than the untreated control cells and ~1.5-fold higher than the unmodified Vun100bv nanobody (Fig. 5C). Finally, we assessed if PCTD-Vun100bv drives full lytic infection in Kasumi-3 cells, which might pose a severe risk for immunosuppressed patients. We transferred the supernatant of infected and treated Kasumi-3 cells onto fibroblasts and counted IE expression four days later. Apart from the positive control PMA, we did not see a significant increase in viral production in any of the conditions, confirming an absence of lytic virus replication (Fig. 5D). This suggests that the induction of IE gene expression alone does not suffice to drive viral DNA replication and virion production.

Due to dysregulated latency-relevant signalling pathways in Kasumi-3 cells compared to primary myeloid cells, their use as a cell line latency model is limited (Crawford et al, 2022). Therefore, we studied the effect of PCTD-Vun100bv treatment in CD14+ monocytes, a site of HCMV latency in vivo. First, we found that PCTD-Vun100bv promotes US28 degradation to a similar extent as seen in Kasumi-3 cells (Fig. 6A). We then analysed if nanobody treatment induces IE gene expression in HCMV-infected monocytes. Primary CD14+ monocytes were isolated from healthy donors and infected with HCMV-IE2-eYFP. Two hours post-infection, cells were treated with PCTD-Vun100bv, a non-US28-targeting PCTD-αBC2 control, and the positive control PMA. Pleasingly, PCTD-Vun100bv induced ~2 times higher IE expression levels than the non-US28-targeting PCTD-αBC2 control

(Fig. 6B). As expected, cells treated with PMA, our positive control, showed the highest levels of IE expression.

## Discussion

We have shown in a variety of cell types and against a range of targets that PCTD is able to direct internalising receptors to the endolysosomal degradation pathway. At present, the mechanism by which PCSK9 drives LDLR degradation is unclear. Previous work has shown that the PCSK9 C-terminal domain is necessary for LDLR degradation (Zhang et al, 2008), and our data presented here suggest that this domain is sufficient to cause endolysosomal degradation of transmembrane receptors to which it is recruited. It has been suggested that PCTD might drive lysosomal trafficking of LDLR by association with the amyloid precursor protein-like protein-2 (APLP2) (DeVay et al, 2013). APLP2 is itself targeted to endolysosomes, so that LDLR might be co-trafficked with APLP2 as part of an LDLR-PCSK9-APLP2 complex. However, whilst it is plausible that PCSK9 degrades LDLR by co-association with a lysosomally targeted transmembrane protein, subsequent work has suggested that APLP2 is not, in fact, necessary for LDLR degradation by PCSK9 (Butkinaree et al, 2015; Fu et al, 2017); thus, the mechanism by which PCTD degrades target proteins remains unclear. Nonetheless, our data shows that, regardless of the molecular mechanism responsible, target transmembrane proteins must traffic through endosomal compartments in order to be subject to PCTD-mediated degradation. Promisingly, we have demonstrated that this includes therapeutically relevant targets, such as TfR and the HCMV latency-associated protein US28. Since TfR is overexpressed in many human cancers it is an attractive drug target, yet its widespread basal distribution makes off-target effects problematic. Clearance from the cell surface and endolysosomal targeting might increase the efficacy of TfR inhibition or boost TfR-mediated ADC delivery, potentially lowering the clinically effective dose and mitigating against off-target effects. In addition, efficient PACTAC-mediated TfR degradation by antibody fragments has potential advantages over the use of whole IgG due to improved tumour penetration (Deonarain et al, 2018).

Moreover, our findings show that PACTAC could be used as a "shock-and-kill" therapeutic in HCMV infection. HCMV reactivation from latency poses a severe risk to patients undergoing organ transplantation, and whilst such reactivation in transplant recipients can be treated with antivirals that target lytic infection, no antivirals that target latently infected cells exist (Krishna et al, 2019). Previous studies using pan-specific histone deacetylase inhibitors and I-BETs (Bromodomain and extra-terminal

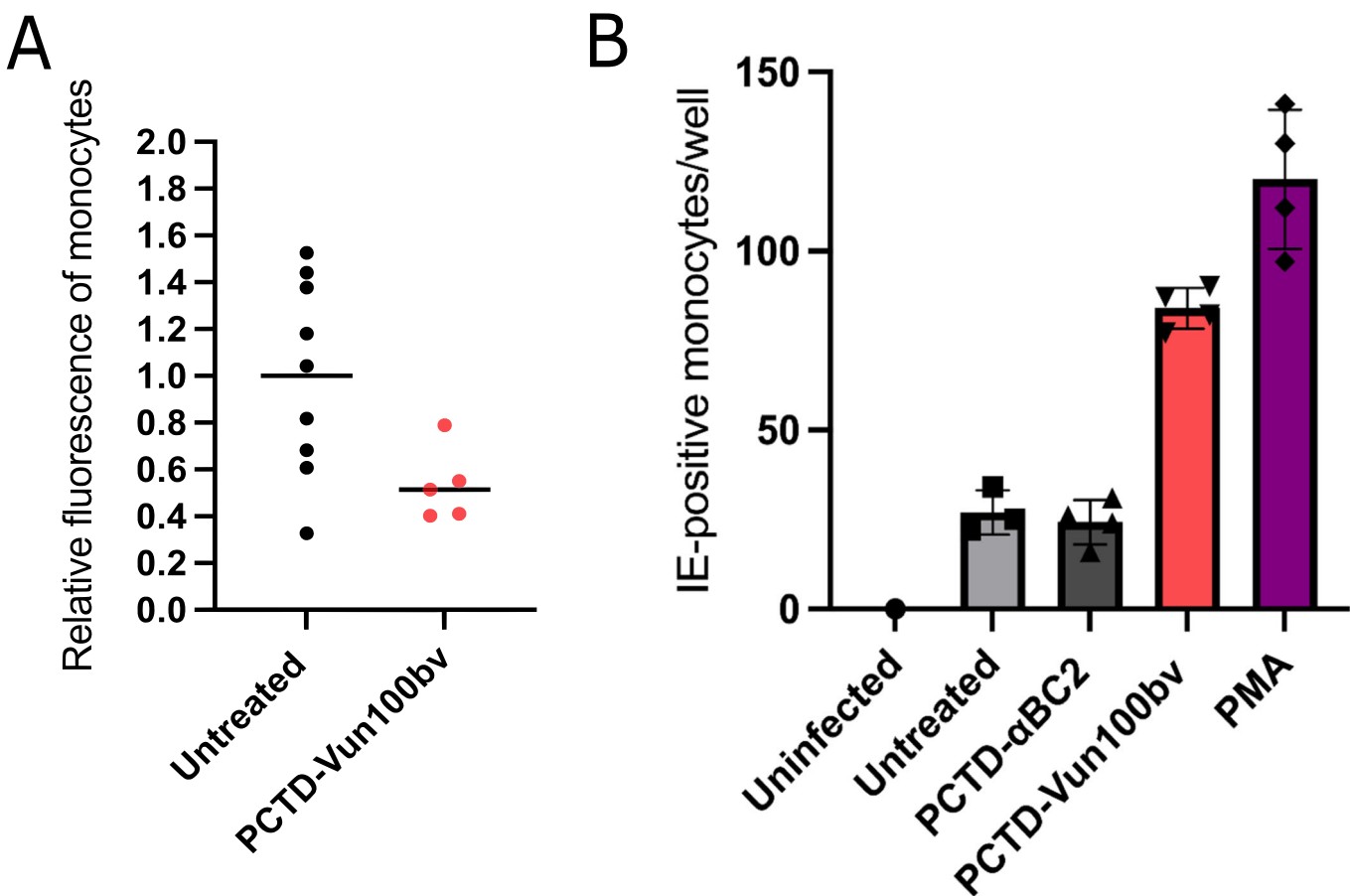

**Figure 6. PCTD-Vun100bv degrades US28 and induces IE gene expression in CD14+ monocytes.**

(A) CD14+ monocytes were infected with HCMV-US28-GFP and were treated with PCTD-Vun100bv or left untreated five days post-infection. The reporter fluorescence (assessed by fluorescence microscopy) was normalized to the average reporter fluorescence of untreated cells. (B) CD14+ monocytes were isolated, infected with HCMV-IE2-eYFP, and were treated with non-US28-targeting PCTD-αBC2, PCTD-Vun100bv, or PMA two hours post-infection. IE-positive nuclei per well were counted two days post-treatment. Data information: Data show individual data points and mean representative of two technical (A) and two biological (B) replicates.

inhibitors) (Groves et al, 2021; Krishna et al, 2016) have shown that the induction of transient IE gene expression in latently infected cells allows T-cell-mediated killing (shock-and-kill) (Krishna et al, 2016; Kim et al, 2018; Nehme et al, 2019). This is due to the extremely high frequency of HCMV-specific cytotoxic T-cells (CTLs) present in normal healthy HCMV carriers (Khan et al, 2002), which recognise lytic antigens but not normally latently infected cells (Wills et al, 2015). However, inhibitors of cellular epigenetic modifiers like I-BETs could have substantial off-target effects. Thus, we developed a virus-specific US28-targeting nano-body, Vun100bv, that induces transient IE gene expression and allows the killing of latently infected CD14+ monocytes by endogenous CTLs in HCMV seropositive carriers (De Groof et al, 2021). In this study, we modified Vun100bv by fusing it to the C terminus of PCTD and show that PCTD-Vun100bv is able to drive US28 degradation. Notably, it induces higher levels of IE gene expression than its precursor molecule, Vun100bv, in the absence of full lytic viral replication. The unmodified Vun100bv nanobody (De Groof et al, 2021) exhibits a similar phenotype, in that it stimulates IE gene expression while restricting full viral reactivation for reasons that are, so far, unclear. It is well established that

infection of CD34+ progenitors (Humby and O'Connor, 2015b) and CD14+ monocytes (Krishna et al, 2017) with HCMV in which US28 has been deleted not only results in IE gene expression, but also in the production of infectious virus. One interpretation of this could be that the presence of US28 at the earliest times of infection (as an incoming viral structural protein) establishes a cell phenotype that not only represses IE expression but also helps suppress Early (E) and Late (L) gene expression. In contrast, removing US28 expression once latency has been established derepresses IE expression but not E and L promoter activity. Alternatively, the incomplete inhibition of US28 activity by Vun100bv and incomplete degradation by PCTD-Vun100bv could cause the inability of these nanobodies to induce lytic infection. While further in vitro studies and clinical trials will be necessary to establish the full therapeutic benefits of PCTD-Vun100bv, our study suggests that, when employed as a shock-and-kill therapeutic, PCTD-Vun100bv may be able to meaningfully lower the HCMV burden in the graft and reduce the risk of HCMV recurrence post-transplant.

More generally, PCTD is a comparatively small domain (~25 kDa) that is independently folded and can be genetically

fused to antibody domains for simple recombinant expression in mammalian cells. As an endogenous human-secreted protein, PCTD itself is unlikely to elicit a strong immune response. Together, these properties make PCTD attractive as a means of targeting antibody-bound receptors for degradation, expanding the range of technologies that have been developed for this purpose (VanDyke et al, 2022). Notably, PACTAC is effective at driving degradation when fused to single-domain antibodies or Fab fragments. Antibody fragments have intrinsic disadvantages as therapeutics: their small size results in rapid renal clearance (Li et al, 2017) and, lacking an Fc region, they do not undergo FcRn-mediated recycling; these two factors combine to result in short serum half-lives (Jin et al, 2022). Nonetheless, antibody fragments potentially have greater solid tumour penetration than full-size IgG, lack Fc-mediated bystander effects, and can be modified to extend serum half-life, for example, by PEGylation (Jin et al, 2022). PCTD fusion to antibody fragments represents an attractive way of combining these advantageous features with targeted degradation.

Our data suggest that PCTD acts in the endosomal system to divert antibody-bound targets towards endolysosomal compartments. Thus, targets that are amenable to PACTAC will be primarily those that undergo constitutive internalisation and recycling to the cell surface. Notably, a wide range of therapeutic antibody targets undergo continuous recycling in this manner, including targets of ADC therapeutics such as Erbb2 and CD33—this recycling is likely to have a negative impact on the effectiveness of ADCs against these targets, due to reduced exposure to endolysosomal compartments (Hammood et al, 2021), but makes them promising targets for PACTAC. We thus envisage PACTAC as a useful alternative to other targeted membrane protein degradation technologies.

# Methods

## Reagents and tools

See Table 1.

## Cell culture and virus infection

All cells were incubated at 37 °C in 5% $CO_2$. Human fetal foreskin fibroblasts (Hfff2, ECACC 86031405) were cultured in Dulbecco's Modified Eagle Medium (DMEM, Merck) supplemented with sterile-filtered 10% heat-inactivated foetal bovine serum (FBS; PAN Biotech) and penicillin (100 U/ml) plus streptomycin (100 µg/ml) (Merck). Primary CD14+ monocytes were maintained in X-Vivo 15 (Lonza) with 2.5 mM additional L-glutamine (Gibco). Kasumi-3 cells (ATCC, CRL-2725) were grown in Roswell Park Memorial Institute (RPMI) medium-1640 (Merck), with sterile-filtered 20% heat-inactivated FBS (PAN Biotech) and penicillin (100 U/ml)/streptomycin (100 µg/ml) (Merck). Hfff2 and Kasumi-3 cells were routinely tested for mycoplasma during maintenance. SK-BR-3 cells (C0006007, AddexBio/Caltag Medsystems) and HeLa cells (C0008001, AddexBio/Caltag Medsystems) were STR-profiled by the suppliers immediately prior to use and were cultured in Dulbecco's Modified Eagle Medium (DMEM, Merck) supplemented with 10% sterile-filtered foetal bovine serum (FBS; Merck) and penicillin (100 U/ml) plus streptomycin (100 µg/ml) (Merck).

**Table 1. Reagents and tools.**

| Reagent/resource | Reference or source | Identifier or Catalog number |
|---|---|---|
| **Experimental models** | | |
| HEK293 F cells | ThermoFisher/Gibco | R79007 |
| Hfff2 cells | ECACC | 86031405 |
| Primary CD14+ monocytes | This study | N/A |
| Kasumi-3 cells | ATCC | CRL-2725 |
| SK-BR-3 cells | AddexBio/Caltag Medsystems | C0006007 |
| HeLa cells | AddexBio/Caltag Medsystems | C0008001 |
| *E. coli* BL21(DE3)pLysS cells | ThermoFisher/Invitrogen | C606010 |
| HCMV-TB40/E-IE2-eYFP | Weekes et al, 2013 | N/A |
| HCMV-TB40/E-US28-GFP | Gift from Thomas Stamminger | N/A |
| HEK 293ET cell | Gift from Paul Manna | N/A |
| **Recombinant DNA** | | |
| PCTD (PCSK9 residues 450 to 692) | This study | Uniprot Q8NBP7 |
| pLXIN2-TfR-GFP | This study | N/A |
| pBA-TfR-GFP | Addgene | 45060 |
| pHLSec | Addgene | 99845 |
| pHLSec-PCTD-αBC2 | This study | N/A |
| pHLSec-PCTD-αGFP | This study | N/A |
| pHLSec-PCTD-Vun100bv | This study | N/A |
| pHLSec-OKT9VL-CL | This study | N/A |
| pHLSec-OKT9VH-CH1-PCTD | This study | N/A |
| pHLSec-OKT9VH-CH1-2HA | This study | N/A |
| Super PiggyBac transposase vector | System Biosciences | PB210PA-1 |
| pB-BC2-CD8a-NPxY-EGFP | This study | N/A |
| pB-BC2-CD8a-ΔMOTIF-EGFP | This study | N/A |
| pB-BC2-CD8a-TGN-EGFP | This study | N/A |
| pMW-αBC2 | This study | N/A |
| pMW-αGFP | This study | N/A |
| Vun100bv | De Groof et al, 2021 | Gift from Martine Smit |
| **Antibodies** | | |
| Anti-TfR rabbit monoclonal | Abcam | EPR4013 |
| Anti-GFP rabbit polyclonal | Abcam | ab6556 |
| Anti-actin rabbit polyclonal | Merck/Sigma | A2066 |
| Goat Anti-Rabbit IgG H&L (Alexa Fluor 680) preadsorbed | Abcam | ab186696 |
| APC-labelled mouse monoclonal anti-CD8 α chain | eBioscience | RPA-T8 |
| **Chemicals, enzymes and other reagents** | | |
| Heat-inactivated foetal bovine serum (FBS) | PAN Biotech | P30-3306 |
| Foetal bovine serum (FBS) | Merck | |

**Table 1.** (continued)

| Reagent/resource | Reference or source | Identifier or Catalog number |
|---|---|---|
| Dulbecco's Modified Eagle Medium (DMEM) | Merck | D5796 |
| Penicillin / Streptomycin | Gibco | 15150 |
| X-Vivo 15 | Lonza | 02-053Q |
| L-glutamine | Gibco | 25030081 |
| Roswell Park Memorial Institute (RPMI) medium-1640 | Merck | R8758 |
| Dulbecco's Phosphate-Buffered Saline (PBS) | Merck | D8662 |
| TransIT-HeLaMONSTER kit | Mirus | MIR2900 |
| NiNTA agarose | Qiagen | 30210 |
| Freestyle HEK media | ThermoFisher/Gibco | 12338018 |
| Accutase | Merck | SCR005 |
| 293Fectin | ThermoFisher | 12347019 |
| Bafilomycin A1 | Merck | SML1661 |
| CompBeads Anti-Mouse Ig | Becton Dickinson | 552843 |
| phorbol 12-myristate 13-acetate (PMA) | Sigma | 16561-29-8 |
| Lymphoprep | STEMCELL Technologies | 7801 |
| Dulbecco's Phosphate-Buffered Saline (PBS) without Calcium and Magnesium | Merck | D8537 |
| Heparin sodium | Wockhardt | PL 29831/0111 |
| CD14 MicroBeads | Miltenyi Biotec | 130-050-201 |
| Trypsin-EDTA | Gibco | 15400054 |
| **Software** | | |
| Fiji/ImageJ | https://imagej.net/software/fiji/ | |
| FlowJo | Becton Dickinson | |
| Prism v5 | GraphPad | |
| Prism v9.5.0 | GraphPad | |
| Prism v10.1.0 | GraphPad | |
| BD Accuri C6 software v1.0.264.21 | BD Biosciences | |
| **Other** | | |
| FACSMelody | Becton Dickinson | |
| LSR Fortessa | Becton Dickinson | |
| BD Accuri C6 Plus flow cytometer | BD Biosciences | |
| Cellomics ArrayScan XTI | ThermoFisher | |
| TE200 microscope | Nikon | |

## Vectors

The gene encoding human transferrin receptor tagged at the C-terminus with EGFP (pLXIN2-TfR-GFP) was amplified from the vector pBA-TfR-GFP (a gift from Gary Banker & Marvin Bentley (Addgene plasmid # 45060; http://n2t.net/addgene:45060;

RRID:Addgene_45060))(Burack et al, 2000) using primers that appended BamHI and NotI restriction sites and subcloned via these sites into a modified pLXIN2 vector (gift from Paul Manna, University of Cambridge). Genes encoding PCTD-nanobody fusions (PCTD-αBC2, PCTD-αGFP and PCTD-Vun100bv), and the genes encoding the OKT9 Fab fragments (Ferrero et al, 2021) $V_L$-$C_L$, $V_H$-$C_{H1}$-PCTD and $V_H$-$C_{H1}$-2HA, were obtained as synthetic DNA blocks (gBlocks; Integrated DNA Technologies) with in-frame AgeI/KpnI restriction sites (except for OKT9 $V_L$-$C_L$, which has a stop codon before the KpnI site) and subcloned into the vector pHLSec (a gift from Edith Yvonne Jones (Addgene plasmid # 99845; http://n2t.net/addgene:99845; RRID:Addgene_99845))(Aricescu et al, 2006) via these sites to yield secreted constructs with C-terminal hexahistidine tags (except for the OKT9 $V_L$-$C_L$ fragment). The PCTD fragment encompasses residues 450 to 692 of PCSK9 (Uniprot Q8NBP7). PCTD-αBC2 and αBC2 were cloned with an in-frame C-terminal Myc tag immediately before the hexahistidine tag. In the OKT9 $V_H$-$C_{H1}$-PCTD construct, a 35-residue glycine-serine linker (sequence: GGGGSGGGGSGGGGS GGGGSGGGGSGGGGSGGGSG) separates the C-terminus of $V_H$-$C_{H1}$ from PCTD.

Genes encoding the BC2-CD8α-tagged *NPxY*, *ΔMOTIF* and *TGN* reporter constructs were obtained as synthetic DNA blocks (gBlocks; Integrated DNA Technologies), cloned into pHLSec via the AgeI-KpnI sites, and finally subcloned into modified PB-T PiggyBac cargo vectors (Halo-APP-mNeonGreen, (Januário et al, 2022), a gift from David Gershlick, University of Cambridge) via XbaI-KpnI sites such that the resultant reporter constructs are under the control of a constitutive chicken β-actin promoter (with CMV enhancer and chicken β-actin/ rabbit β-globin chimeric intron) flanked by PiggyBac Transposase recognition sequences. Genes encoding the nanobodies αBC2 and αGFP were obtained as gBlocks (Integrated DNA Technologies) and subcloned into an in-house *E. coli* expression vector, pMW (Owen and Evans, 1998). The invariant BC2-CD8α-transmembrane domain sequence is as follows:

MGILPSPGMPALLSLVSLLSVLLMGCVAETGASMPDRKAAV SHWQQRSGSSQFRVSPLDRTWNLGETVELKCQVLLSNPTSGC SWLFQPRGAAASPTFLLYLSQNKPKAAEGLDTQRFSGKRLGDT FVLTLSDFRRENEGYYFCSALSNSIMYFSHFVPVFLPAKPTTTPA PRPPTPAPTIASQPLSLRPEACRPAAGGAVHTRGLDFACDTRE KKPSSVRALSIVLPIVLLVFLCLGVFLLW

The variable cytoplasmic sequences inserted between the transmembrane domain and GFP are based on the LDLR cytoplasmic tail sequence and are as follows, trafficking motifs underlined:

*NPxY*: KNWRLKNINSINFD<u>NPVY</u>QKTTEDEVHICHNQDGY-SYPSRQMVSLEDDVALESGS

*ΔMOTIF*: KNWRLKNINSINFDAAVAQKTTEDEVHICHNQD GASAPSRQMVSLESGS

*TGN*: KNWRLKNINSINFDAAVAQKTTEDEVHICHNQDGA-SAD<u>YQRL</u>NGASGASLESGS

In each construct the variable cytoplasmic sequence is followed immediately by EGFP (VSKG…DELYK). The assembled Piggybac vectors are pB-BC2-CD8a-*NPxY*-EGFP, pB-BC2-CD8a-*ΔMOTIF*-EGFP and pB-BC2-CD8a-*TGN*-EGFP.

## Piggybac construction of BC2-CD8α reporter cell lines

Cargo vectors encoding the *NPxY*, *ΔMOTIF* and *TGN* reporter constructs under the control of a constitutive chicken β-actin

promoter (with CMV enhancer and chicken β-actin/rabbit β-globin chimeric intron) flanked by PiggyBac Transposase recognition sequences was mixed with Super PiggyBac transposase vector (System Biosciences) in a 5:1 mass ratio (cargo:transposase) and transfected into HeLa cells (grown in a six-well dish) using a TransIT-HeLaMONSTER (Mirus) transfection kit according to the manufacturer's protocol. After 48 h the cells were dissociated with Accutase and plated onto a T75 tissue culture flask. Once near confluence the cells were dissociated with Accutase, pelleted by centrifugation at 300 g, the pellet washed with PBS and gently resuspended in PBS supplemented with 1% FBS, and the cells sorted for GFP fluorescence on a Becton Dickinson FACSMelody sorter.

## Construction of TfR-GFP reporter cell line by retroviral transduction

Transient DNA transfections were carried out using TransIT-HeLaMONSTER kit (Mirus) following the manufacturer's instructions. For stable cell line generation using the pLXIN vectors, HEK 293ET cells were co-transfected with the appropriate retroviral vector and the packaging plasmids pMD.GagPol and pMD.VSVG (gifts from Paul Manna, University of Cambridge) in the ratio of 50:30:15. Virus-containing supernatant was filtered through a 0.45 μm filter and applied directly to the target cells. The cells were subsequently sorted for GFP fluorescence on a Becton Dickinson FACSMelody sorter.

## HEK F expression of proteins by transient transfection

pHLSec vectors containing PCTD-nanobody fusions or OKT9 fab fragments were transfected into 30 ml HEK-293F cells (Thermo Fisher) at a density of $1 \times 10^6$ cells/ml with 30 μg DNA using 293Fectin (Thermo Fisher), following the manufacturer's instructions. In the case of the OKT9 fab fragments, 15 μg of each vector were mixed and transfected in the same way. After 72 h supernatants were harvested, passed through a 0.4 μm filter, bound to NiNTA agarose (Qiagen), washed with Wash buffer (20 mM Tris pH 8, 300 mM NaCl), eluted with Wash buffer supplemented with 300 mM imidazole pH 8, concentrated and dialyzed against PBS. Aliquots were flash frozen in liquid nitrogen.

## *E. coli* expression of nanobodies

pMW vectors encoding the αBC2 and αGFP nanobodies were expressed in *E. coli* strain BL21(DE3)pLysS and purified by NiNTA affinity chromatography as for the HEK F-expressed proteins.

## Antibodies

Anti-GFP (Abcam, ab6556, rabbit polyclonal), Anti-actin (Sigma, A2066, rabbit polyclonal), Goat Anti-Rabbit IgG H&L (Alexa Fluor 680) preadsorbed (Abcam, ab186696), APC-labelled mouse monoclonal anti-CD8 a chain, RPA-T8 (eBioscience), Anti-TfR (Abcam, RabMAb EPR4013).

## Cell treatment procedure

For Figs. 1, 3, 4, cells were seeded in 6-well dishes (Falcon) at ~$0.3 \times 10^6$ cells/well in DMEM supplemented with 10% FBS,

penicillin (100 Units/ml) and streptomycin (100 μg/ml) and grown for 24 h; for Fig. 2, SK-BR-3 cells were seeded in 6-well dishes at $0.4 \times 10^6$ cells/well (in the same medium). The medium was aspirated and replaced with 1 ml fresh DMEM supplemented as above, and protein treatments added in equal volumes of Dulbecco's PBS; for 'untreated' cell samples, an equal volume of Dulbecco's PBS was added. 5 μg/ml concentrations of PCTD-nanobody fusions were used unless otherwise indicated, or a molar equivalent (120 nM) of the nanobody. 'Overnight' treatments were typically performed for 16 h before cells were lysed. Where used, Bafilomycin A1 (or an equal volume of DMSO) was added at 100 nM 2 h prior to treatment with PCTD fusions or control antibodies.

## Western blot analysis

Adherent cells growing in 6-well dishes were dissociated with Accutase, collected into microcentrifuge tubes and pelleted at $300 \times g$ for 5 min. The supernatants were aspirated and the pellets frozen on dry ice. The pellets were then lysed by resuspension in NP40 lysis buffer (50 mM HEPES pH 7.4, 150 mM NaCl, 1% NP40) and incubation on ice for 30 min with occasional mixing by vortexing. The suspension was then centrifuged at $7000 \times g$ for 3 min at 4 °C and the supernatants retained. Total protein concentration was measured using a BCA assay (Pierce) and typically 40 μg of protein was loaded on a Bio-Rad TGX 4–20% SDS-PAGE gel and blotted using a Bio-Rad wet transfer system onto Amersham Protran nitrocellulose membranes for 2 h at constant voltage (70 V). The membranes were blocked for 1 h at 21 °C with PBS + 5% skimmed milk powder (Marvel). Anti-GFP, anti-actin and anti-TfR were used at 1:1000 dilutions into PBS + 5% skimmed milk powder + 0.1% Tween-20, and incubated with the blot overnight at 4 °C. Blots were washed three times with PBS + 0.1% Tween-20 (5 min per wash) before addition of goat Anti-Rabbit IgG H&L (Alexa Fluor 680) diluted 1:1000 in PBS + 5% skimmed milk powder + 0.1% Tween-20 and incubation for 1 h at 21 °C. Blots were washed three times as above and imaged using a Li-Cor Odyssey near-infrared fluorescence imaging system. Blot bands were quantitated by densitometry using Fiji.

## Flow cytometry

Data for Figs. 3, 4, and EV1 was collected on a Becton Dickinson LSR Fortessa and analysed with FlowJo software. GFP fluorescence was excited with the 488 nm laser and measured with the 530/30 nm bandpass filter. APC fluorescence was excited with the 640 nm laser and measured with the 670/14 nm bandpass filter. For the anti-CD8 uptake experiment, compensation was performed using Becton Dickinson CompBeads (following the manufacturer's instructions) with APC-labelled anti-CD8 alpha chain, and untreated *TGN* cells (for GFP fluorescence), and applied with the FlowJo software. Where applicable, transduced HeLa cells were sorted on a Becton Dickinson FACSMelody according to the manufacturer's protocols. Briefly, for sorting of stably transfected GFP-tagged reporter cell lines, nonfluorescent cells were excluded using a gate set to exclude the whole of a control population of unmodified HeLa cells.

For the US28 degradation assay via flow cytometry (Fig. EV4), US28 expression was assessed by measuring the GFP intensity of PCTD-Vun100bv-treated fibroblasts. To do this, fibroblasts were

infected with HCMV-TB40/E-US28-GFP. Three days post-infection, cells were treated with PCTD-Vun100bv (500 nM), Vun100bv (500 nM), or were left untreated. Two days post-treatment, adherent fibroblasts were washed with PBS, dissociated with Trypsin-EDTA, and transferred to FACS tubes. Cells were analyzed for GFP with a BD Accuri C6 Plus flow cytometer and BD Accuri C6 software (version 1.0.264.21).

## Anti-CD8 feeding experiment

Reporter cells, and unmodified HeLa as a negative control, were seeded in six-well dishes (in 1 ml DMEM + 10% FBS + penicillin (100 U/ml) and streptomycin (100 µg/ml)) at a density of $0.6 \times 10^6$ cells/well and grown overnight (to ~70% confluence). The medium was then supplemented with 5 µg APC-labelled anti-CD8 alpha chain and the cells incubated for a further 30 min at 37 °C/5% $CO_2$. Cells were then placed on ice, washed twice with cold PBS, then dissociated with trypsin. The cells were pelleted by centrifugation and washed once with PBS, and then analysed by flow cytometry.

## Virus infection

Monocytes and Kasumi-3 cells were infected with viral isolate RV 1664 (HCMV-TB40/E-IE2-eYFP) (Weekes et al, 2013) or viral isolate HCMV-TB40/E-US28-GFP (kind gift from Thomas Stamminger) at a predicted multiplicity of infection (MOI) of 3. Hfff2 were infected with viral isolate HCMV-TB40/E-US28-GFP at an MOI of 0.1–1.

## Fluorescence microscopy assays

Fibroblasts were infected with HCMV-TB40/E-US28-GFP and were left untreated or were treated with PCTD-Vun100bv (500 nM) or a non-US28-targeting PCTD-αBC2 control (500 nM) three days post-infection. To assess US28 degradation over a time course of 210 min, pictures were taken every 30 min using live-cell microscopy (Cellomics ArrayScan XTI, ThermoFisher), starting 30 min after treatment. The colour of the pictures (EV3a) was inverted to highlight the decrease in GFP intensity over time. The decay in fluorescent intensity signal for each spot was normalised to the mean intensity of each spot measured at the first time point.

Kasumi-3 cells were infected with HCMV-TB40/E-US28-GFP. Two days post-infection, cells were treated with PCTD-Vun100bv (500 nM), or were left untreated. Pictures were taken three days post-treatment. Monocytes infected with HCMV-TB40/E-US28-GFP were treated with PCTD-Vun100bv (200 nM) or were left untreated five days post-infection. Pictures were taken four days post-treatment. For Kasumi-3 cells and monocytes, brightness measurements were performed using ImageJ (version 1.53e). The mean spot intensity of infected cells was measured by selecting representative cells with the freehand selection tool of ImageJ, then subtracting the mean background intensity for each field of view. The measured reporter fluorescence of treated cells was normalized to the average fluorescence of untreated cells.

## Reactivation of Kasumi-3 cells and CD14+ monocytes

Reactivation of Kasumi-3 cells and CD14+ monocytes with phorbol 12-myristate 13-acetate (PMA, Merck) was achieved at a concentration of 20 ng/mL (O'Connor and Murphy, 2012; Poole et al, 2021).

## Isolation of peripheral blood mononuclear cells and selection of CD14+ monocytes

Peripheral blood mononuclear cells (PBMCs) were obtained by venipuncture from healthy volunteers. The whole blood samples were diluted in PBS without calcium and magnesium, supplemented with 100 U/ml heparin sodium (Wockhardt). PBMCs were isolated from whole blood samples by employing a density gradient centrifugation with Lymphoprep (Alere Technologies) solution and harvesting the cells from the interface layer. To separate the CD14+ monocytes from the remaining PBMCs, the harvested cells were incubated in chilled filtered magnetic activated cell sorting (MACS) buffer (PBS without calcium and magnesium, supplemented with 2 mM EDTA). Monocytes were selected by performing MACS using anti-CD14 magnetic beads (Miltenyi Biotec) following manufacturer's instructions. Cells were counted and plated in PBS with calcium and magnesium for one to three hours, at a density of $5 \times 10^5$ cells per cm². After three hours, PBS was replaced with X-Vivo 15 (Lonza) supplemented with 2.5 mM L-glutamine (Gibco).

## Detection of IE expression

Kasumi-3 cells and CD14+ monocytes were seeded in 96-well plates. The following day, the medium was removed and cells were left uninfected or infected with HCMV-RV1164-IE2-eYFP. Two hours post-infection, the medium was replaced with medium either containing PCTD-Vun100bv (500 nM), PMA at a final concentration of 20 ng/ml (positive control), Vun100bv (100 nM, as described previously (De Groof et al, 2021)), non-US28-targeting PCTD-αBC2 (500 nM) or untreated medium (negative control). Two to four days post-infection, IE expression was detected by manually counting for eYFP-expressing cells with a widefield Nikon TE200 microscope. The supernatant of Kasumi-3 cells was transferred onto fibroblasts three to four days post-treatment, and IE2-eYFP-positive cells were counted four days after supernatant transfer.

## General PACTAC usage

A PACTAC experiment comprises several stages as follows.

- A suitable antibody with known sequence is required. The antibody must bind an epitope that is accessible from the cell's exterior.
- Clone the heavy and light chains into separate mammalian expression vectors such as pHLSec, inserting the PCTD domain at the C-terminus of the heavy chain and separated from it by a glycine-serine linker (such as the 35-residue linker used herein). If using a whole IgG, it is likely that it can be left untagged and the PACTAC purified by Protein G affinity chromatography; however, we have not yet tested this approach in our laboratory. Alternatively, clone the heavy chain, or heavy chain fragment, with a C-terminal hexahistidine tag.
- Express the PACTAC in 30 mL HEK 293 F cells by transfection with 293Fectin transfection reagent as follows. Use a ratio of 1 µg total DNA per $1 \times 10^6$ cells. For coexpression of heavy and light chains of an IgG, use an equal mass of each plasmid; this ratio may need to be optimised to improve expression yield.
- Cells should be at a density of ~ $2 \times 10^6$ cells per mL so that upon

splitting the cells for transfection at $1 \times 10^6$ cells per mL, about half the media is replaced with fresh media. Then prepare a mixture of plasmid DNA totalling 30 μg and follow the 293Fectin reagent protocol as detailed by the manufacturer.

- After 3 days, harvest the cells by centrifugation. Pass the supernatant through a 0.45 μm filter to exclude any remaining cells. Add protease inhibitors to the clarified supernatant and proceed with affinity chromatography (e.g. using Ni-NTA agarose) following standard protein purification protocols.
- Plate adherent cells expressing the target protein in six-well dishes in an appropriate growth medium at a density that allows treatment for the desired time period without overgrowth of the cells.
- Add PACTAC directly to the culture medium at defined concentrations, gently swirl to ensure mixing of the PACTAC with the medium, and incubate the cells for the desired time period.
- Aspirate culture medium, wash with PBS and harvest the cells as appropriate for the desired assay to determine effects on the level of the target protein.

## Statistical analysis

Statistical analysis was performed using GraphPad Prism (v.5 for Figs. 1, 3 and EV1; v9.5.0 and v10.1.0 for Figs. 5, 6, EV3, EV4). Data are plotted as mean ± SD. The type of statistical analysis is indicated in the figure legend. When using statistical methods based on the normal distribution, data was tested for normality using Shapiro-Wilk test.

## Curve fitting

Curve fitting (Fig. 2) was performed using Prism 5.

# Data availability

This study includes no data deposited in external repositories.

# Peer review information

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

## Acknowledgements

This research was supported by the CIMR Flow Cytometry Core Facility. In particular, we wish to thank Gabriela Grondys-Kotarba and Reiner Schulte for their advice and support in flow cytometry and cell sorting. We also wish to thank Nika Romashova for her support in Cellomics microscopy. We would also like to thank Raimond Heukers, Timo De Groof, and Marine Smit for their advice on Vun100bv. BTK and DJO were supported by a Wellcome Trust Principal Research Fellowship to DJO (WT 207455/Z/17/Z). This work was supported by the Medical Research Council (grant numbers MR/S00081X/1) Programme Grant to J Sinclair and the Cambridge NIHR BRC Cell Phenotyping Hub. J Schmitt was supported by an MD Fellowship of the German Center for Infection Research (DZIF MD programme TI 07.003). SCG was supported by a Sir Henry Dale Fellowship, jointly funded by the Wellcome Trust and the Royal Society (098406/Z/12/B). For the purpose of open access, the authors have applied a Creative Commons Attribution (CC BY) license to any Author Accepted Manuscript version arising from this submission.

## Author contributions

**Janika Schmitt**: Conceptualization; Formal analysis; Investigation; Writing—original draft; Writing—review and editing. **Emma Poole**: Conceptualization; Formal analysis; Investigation. **Ian Groves**: Conceptualization; Formal analysis; Investigation. **David J Owen**: Conceptualization; Funding acquisition; Writing—review and editing. **Stephen C Graham**: Conceptualization; Formal analysis; Investigation; Writing—original draft; Writing—review and editing. **John Sinclair**: Conceptualization; Funding acquisition; Writing—original draft; Writing—review and editing. **Bernard T Kelly**: Conceptualization; Formal analysis; Investigation; Writing—original draft; Writing—review and editing.

## Disclosure and competing interests statement

The University of Cambridge has filed a patent application related to this work.

## Ethical approval

All research describing studies on primary human material with HCMV was assessed and approved by the Cambridge Local Research Ethics committee (reference 97/092). Informed consent was received from blood donors with the Cambridge Local Research Ethics committee and the Cambridge Internal Review Board. Cells were harvested from healthy adult donors.

# Expanded View Figures

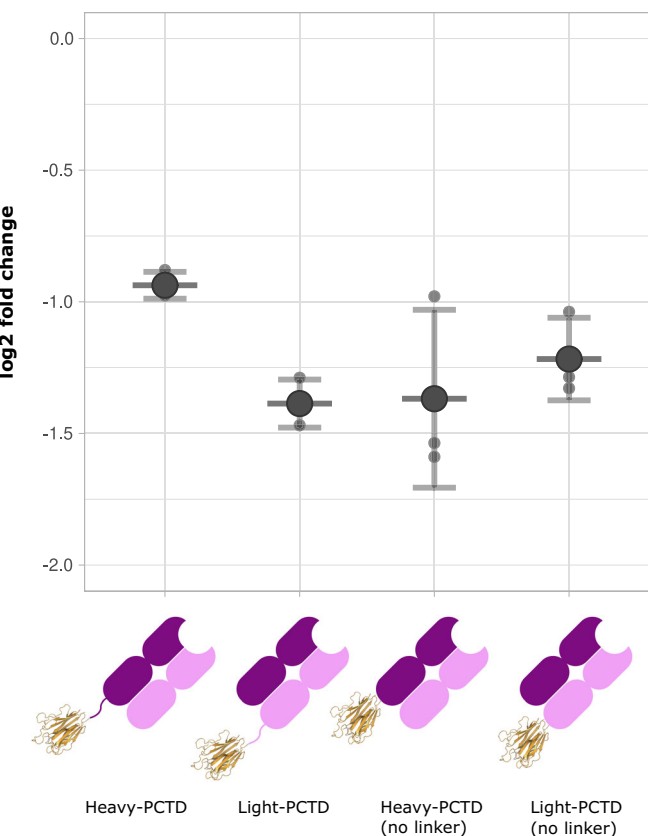

Heavy-PCTD  Light-PCTD  Heavy-PCTD  Light-PCTD
(no linker)  (no linker)

**Figure EV1. PCTD fusion format does not significantly affect efficiency of OKT9FabPCTD degradation.**

SK-BR-3 cells were treated with 25 nM of four different OKT9FabPCTD fusions and TfR degradation measured by Western Blot and densitometry and shown as log(2)-fold change. The different fusion formats are depicted as cartoons, with OKT9 heavy chains shown in dark purple, light chains shown in light purple, and PCTD shown in gold as a molecular model. 35 amino acid glycine/serine linkers are depicted as solid lines separating OKT9 from PCTD. 3 biological replicates per sample are plotted as separate data points together with mean and s.e.m. One-way ANOVA indicates no significant differences ($p > 0.05$) between the different formats.

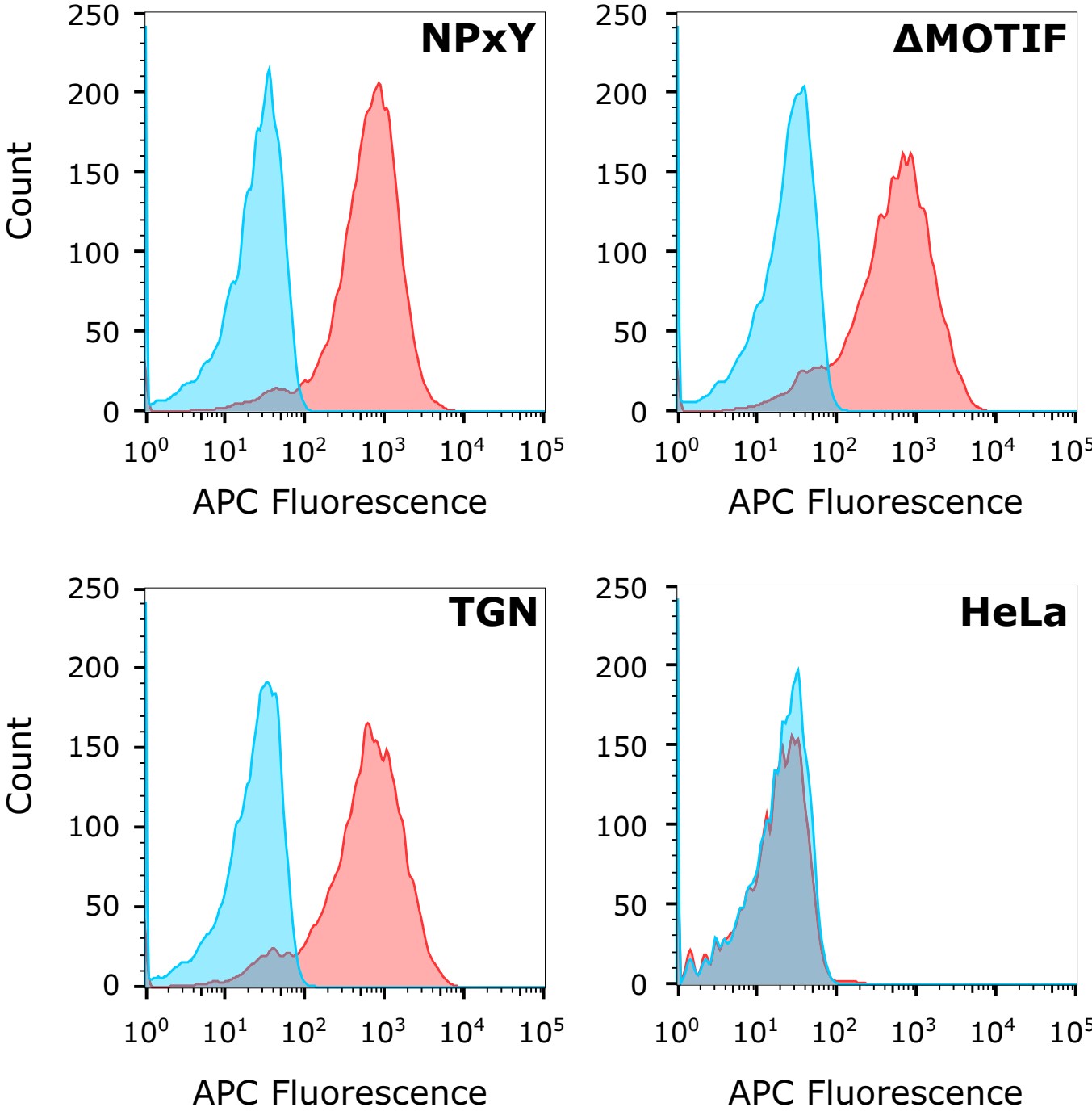

**Figure EV2.   Reporter cell lines can bind and/or take up fluorescent CD8 antibody.**

Reporter cells, or HeLa cells as a negative control, were incubated with an allophycocyanin (APC)-labelled anti-CD8 α chain antibody for 30 min or left untreated, then washed in PBS and harvested for flow cytometric analysis. Because the reporter cell lines are fluorescent due to the presence of GFP, compensation was performed using Becton Dickinson CompBeads. Traces are depicted for a representative sample from two biological replicate experiments; red traces depict cells exposed to anti-CD8 antibody and blue traces depict untreated cells. All reporter cell lines display similar levels of anti-CD8 antibody binding/uptake, whereas HeLa cells do not take up the antibody.

## A

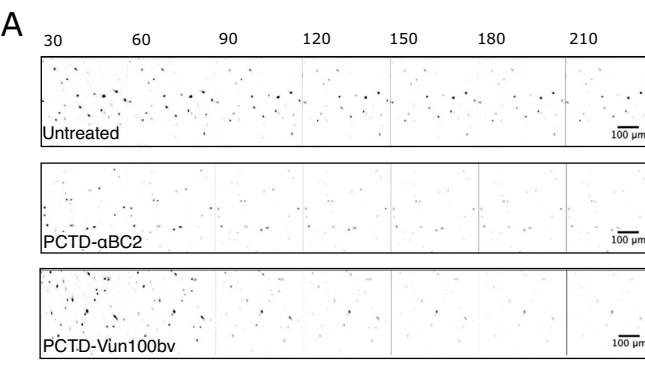

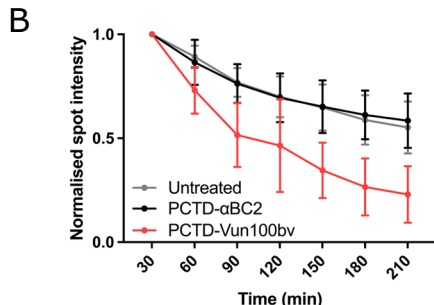

## B

**Figure EV3. PCTD-Vun100bv induces US28 degradation in fibroblasts, whereas non-US28-targeting PCTD-αBC2 does not affect US28 levels.**

(**A**, **B**) Fibroblasts were infected with HCMV-US28-GFP. Three days post-infection, cells were treated with PCTD-Vun100bv, with a non-US28-targeting PCTD-αBC2 control, or left untreated, and then imaged using live-cell fluorescence microscopy. (**A**) Time course series generated with live-cell fluorescence microscopy. Numbers indicate minutes post-treatment, starting with 30 min post-treatment. Scale bars: 100 μm. (**B**) Quantified normalised change in per-spot intensity of fluorescence signal from (**A**). A two-way ANOVA with Tukey's test shows a highly significant difference from 60–210 min between PCTD-Vun100b and untreated ($P < 0.0001$), but no significant difference between control and untreated. Data is shown as mean ± SD and is representative of at least 5 technical replicates per condition.

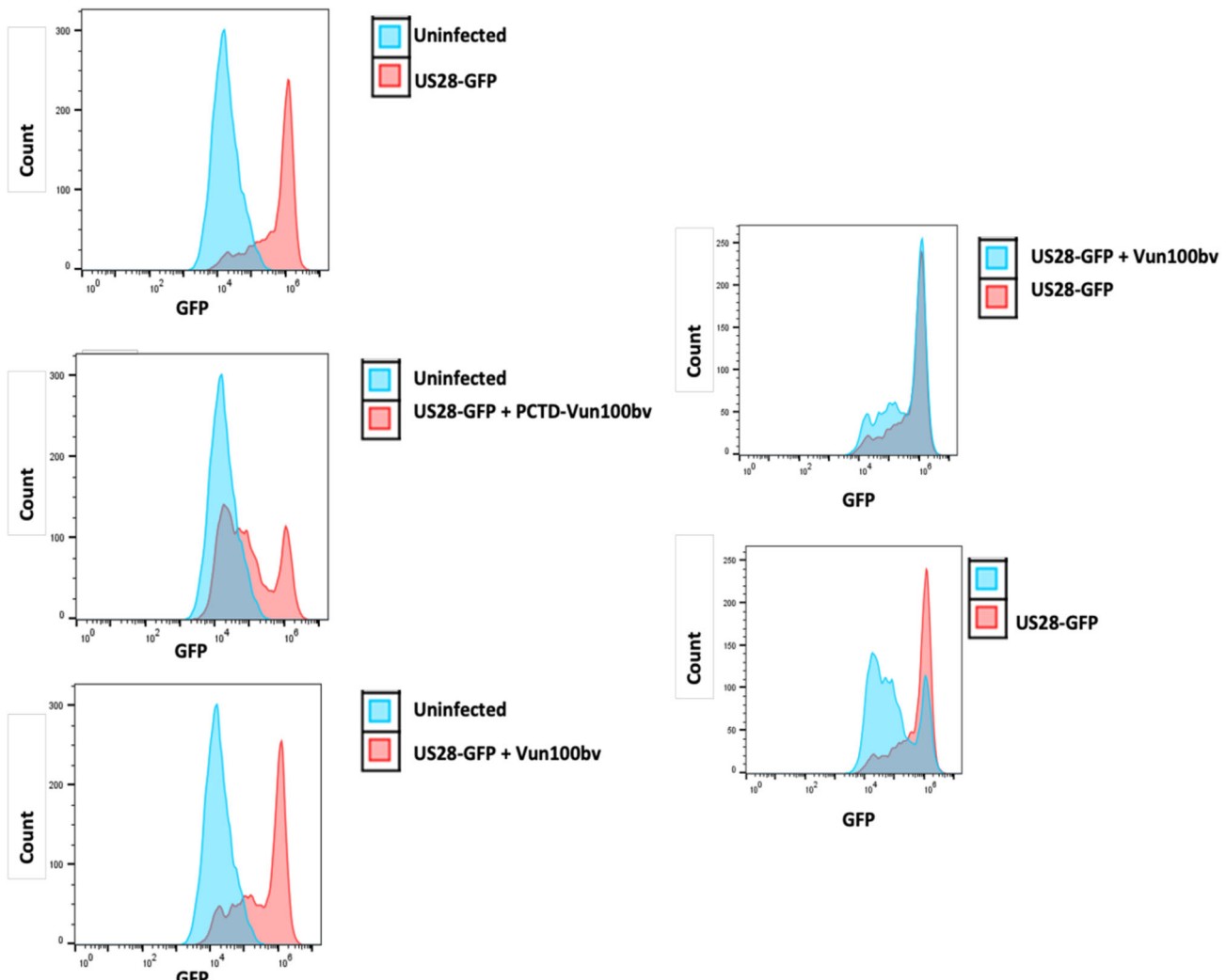

**Figure EV4.   PCTD-Vun100bv degrades US28 in fibroblasts, whereas Vun100bv treatment does not affect US28 levels.**

Fibroblasts were infected with HCMV-US28-GFP and were left untreated or were treated with Vun100bv or PCTD-Vun100bv three days post-infection. Total US28 levels were quantified by measuring fluorescent GFP intensity via flow cytometry. The data is representative of two biological replicates.

                                                       