## [Peer Review File · EMBO Reports]

Repurposing an endogenous degradation domain for antibody-mediated disposal of cell-surface proteins

Janika Schmitt, Emma Poole, Ian Groves, David Owen, Stephen Graham, John Sinclair, and Bernard Kelly
DOI: 10.15252/embr.202357890

Corresponding authors: Bernard Kelly (btk1000@cam.ac.uk) , David Owen (djo30@cam.ac.uk), Stephen Graham (scg34@cam.ac.uk), John Sinclair (js152@cam.ac.uk)

Review Timeline:

Submission Date:	27th Jul 23
Editorial Decision:	29th Aug 23
Revision Received:	16th Nov 23
Editorial Decision:	1st Dec 23
Revision Received:	14th Dec 23
Accepted:	21st Dec 23

Transaction Report:

Dear Dr. Kelly

Thank you for the submission of your research manuscript to our journal. We have now received the full set of referee reports that is copied below.

As you will see, the referees acknowledge that the findings are potentially interesting, but they also provide feedback on how to further strengthen your study, which should be taken on board during the revision. Please address all referee concerns with the exception of identifying co-receptor(s) for the PCTD (Ref 3, point 1), which is not mandatory from our side. While we agree that this would be a very interesting and significant addition, we also recognize the amount of time and effort that would be required.

Please revise your manuscript with the understanding that the referee concerns (as detailed above and in their reports) must be fully addressed and their suggestions taken on board. Please address all referee concerns in a complete point-by-point response. Acceptance of the manuscript will depend on a positive outcome of a second round of review. It is EMBO Reports policy to allow a single round of revision only and acceptance or rejection of the manuscript will therefore depend on the completeness of your responses included in the next, final version of the manuscript.

We realize that it is difficult to revise to a specific deadline. In the interest of protecting the conceptual advance provided by the work, we recommend a revision within 3 months (November 29). Please discuss the revision progress ahead of this time with the editor if you require more time to complete the revisions.

I am also happy to discuss the revision further via e-mail or a video call, if you wish.

You can either publish the study as a short report or as a full article. For short reports, the revised manuscript should not exceed 27,000 characters (including spaces but excluding materials & methods and references) and 5 main plus 5 expanded view figures. The results and discussion sections must further be combined, which will help to shorten the manuscript text by eliminating some redundancy that is inevitable when discussing the same experiments twice.

For a normal article there are no length limitations, but it should have more than 5 main figures and the results and discussion sections must be separate.

All Materials and Methods need to be described in the main text. We would encourage you to use 'Structured Methods', our new Materials and Methods format. According to this format, the Materials and Methods section should include a Reagents and Tools Table (listing key reagents, experimental models, software and relevant equipment and including their sources and relevant identifiers) followed by a Methods and Protocols section in which we encourage the authors to describe their methods using a step-by-step protocol format with bullet points, to facilitate the adoption of the methodologies across labs. More information on how to adhere to this format as well as downloadable templates (.doc or .xls) for the Reagents and Tools Table can be found in our author guidelines: < <https://www.embopress.org/page/journal/14693178/authorguide#manuscriptpreparation>>. An example of a Method paper with Structured Methods can be found here: <<https://www.embopress.org/doi/10.15252/msb.20178071>>.

Please remove the section "The Paper Explained" as it is not part of manuscripts published in EMBO Reports.

*****IMPORTANT NOTE:

We perform an initial quality control of all revised manuscripts before re-review. Your manuscript will FAIL this control and the handling will be delayed IN CASE the following APPLIES:

- 1) A data availability section providing access to data deposited in public databases is missing. If you have not deposited any data, please add a sentence to the data availability section that explains that.
- 2) Your manuscript contains statistics and error bars based on $n=2$. Please use scatter blots in these cases. No statistics should be calculated if $n=2$.

When submitting your revised manuscript, please carefully review the instructions that follow below. Failure to include requested items will delay the evaluation of your revision.*****

2) individual production quality figure files as .eps, .tif, .jpg (one file per figure). Please download our Figure Preparation Guidelines (figure preparation pdf) from our Author Guidelines pages <https://www.embopress.org/page/journal/14693178/authorguide> for more info on how to prepare your figures.

4) a complete author checklist, which you can download from our author guidelines (<<https://www.embopress.org/page/journal/14693178/authorguide>>). Please insert information in the checklist that is also reflected in the manuscript. The completed author checklist will also be part of the RPF.

5) Please note that all corresponding authors are required to supply an ORCID ID for their name upon submission of a revised manuscript (<<https://orcid.org/>>). Please find instructions on how to link your ORCID ID to your account in our manuscript tracking system in our Author guidelines (<<https://www.embopress.org/page/journal/14693178/authorguide#authorshipguidelines>>)

6) We replaced Supplementary Information with Expanded View (EV) Figures and Tables that are collapsible/expandable online. A maximum of 5 EV Figures can be typeset. EV Figures should be cited as "Figure EV1, Figure EV2" etc... in the text and their respective legends should be included in the main text after the legends of regular figures.

7) Please note that a Data Availability section at the end of Materials and Methods is now mandatory. In case you have no data that requires deposition in a public database, please state so instead of refereeing to the database. See also < <https://www.embopress.org/page/journal/14693178/authorguide#dataavailability>>. Please note that the Data Availability Section is restricted to new primary data that are part of this study.

Additional information on source data and instruction on how to label the files are available <<https://www.embopress.org/page/journal/14693178/authorguide#sourcedata>>.

10) Figure legends and data quantification:
The following points must be specified in each figure legend:

- the name of the statistical test used to generate error bars and P values,
 - the number (n) of independent experiments (please specify technical or biological replicates) underlying each data point,
 - the nature of the bars and error bars (s.d., s.e.m.)
- If the data are obtained from n {less than or equal to} 5, show the individual data points in addition to the SD or SEM.
- If the data are obtained from n {less than or equal to} 2, use scatter blots showing the individual data points.

See also the guidelines for figure legend preparation:
<https://www.embopress.org/page/journal/14693178/authorguide#figureformat>

11) Our journal encourages inclusion of *data citations in the reference list* to directly cite datasets that were re-used and obtained from public databases. Data citations in the article text are distinct from normal bibliographical citations and should directly link to the database records from which the data can be accessed. In the main text, data citations are formatted as follows: "Data ref: Smith et al, 2001" or "Data ref: NCBI Sequence Read Archive PRJNA342805, 2017". In the Reference list, data citations must be labeled with "[DATASET]". A data reference must provide the database name, accession number/identifiers and a resolvable link to the landing page from which the data can be accessed at the end of the reference. Further instructions are available at <<https://www.emboPress.org/page/journal/14693178/authorguide#referencesformat>>.

12) As part of the EMBO publication's Transparent Editorial Process, EMBO Reports publishes online a Review Process File to accompany accepted manuscripts. This File will be published in conjunction with your paper and will include the referee reports, your point-by-point response and all pertinent correspondence relating to the manuscript.

Yours sincerely,

Referee #1:

The paper by Schmitt and colleagues describes a mechanism for inducing the degradation of cell-surface proteins using modified antibodies. Antibodies are expressed with a PCSK9 domain that elicits target degradation via the endolysosomal system. The authors demonstrate convincing degradation of a few targets, including TfR and HCMV protein US28. The work appears sound and convincing. It is well written and easy to follow. The major caveat, about which the authors are candid, is that it is unknown how PCSK9 is directing target degradation. However, this is a new technology that could find applications in biomedical research without a full understanding of its mechanism.

[REDACTED: Referee's comment and the author's response with unpublished data.]

Figure 4 - the results suggest that PACTAC against US28 does not induce virus replication, only IE expression. Is this linked to the apparently low levels of degradation of US28? Would complete US28 abrogation cause lytic infection?

Referee #2:

The authors report a new method for targeted degradation of cell-surface proteins that exploits the natural function of the endogenous protein PCSK9. In this method, cell surface proteins are targeted with a chimeric molecule consisting of target-binding antibody/antibody fragment fused to the C-terminal domain of PCSK9, which they call PACTACs. Although the mechanism of endogenous PCSK9 action is poorly understood, the authors show that their PACTAC molecules cause lysosomal re-routing of membrane proteins that usually recycle through endosomes back to the cell surface. The authors prove the effectiveness of PACTACs against several reporter proteins as well as the endogenous protein Transferrin. Finally, they design a PACTAC against the human cytomegalovirus encoded protein US28 and show that this approach has therapeutic potential for latent HCMV infection.

The key finding of this manuscript is the use of the C-terminal domain of PCSK9 as a degron that can be fused to antibodies or nanobodies to induce lysosomal degradation of cell surface proteins. This finding is significant as it represents a novel addition

to the growing number of methods for targeted protein degradation, which will be of interest to a broad audience. Overall the manuscript is well-written and the findings are well-supported by high-quality experimental data. I recommend publication after the following minor comments are addressed:

Minor comments:

- 1) Figure 1e,f and Figure 4: It would benefit the reader if the authors include illustrations of the PACTACs used in these experiments. This could be structural models similar to what is already shown in Figure 1a,b and 2b or even simple cartoon illustrations. For example, from just the text it is hard to visualize what the OKT9FabPCTD construct might look like.
- 2) Figure S3a: For the labels the authors should include + sign between US28-GFP and the PACTACs to illustrate that these conditions are HCMV-US28-GFP infected cells plus treatment with e.g. PCTD-Vun100bv. The text/numbers on the X and Y axis of these graphs is very small.
- 3) Figure 4: It would benefit non-virologist/immunologist readers if the authors include a schematic here to show how US28 represses IE2, and that PACTAC degradation of US28 would relieve this repression resulting in IE2-positive cells, which in turn triggers T-cell killing.

Referee #3:

Schmitt et al. describe the development of "PACTACs" (PCSK9-Antibody Chimeras for Targeted Clearance) which allow the targeted degradation of internalizing cell surface proteins. There has been substantial recent interest in this field, including descriptions of bispecific antibodies that drive targeted ubiquitination and subsequent degradation (AbTACs/PROTABs) and, similar to this manuscript, molecules that drive increased internalization and endolysosomal degradation via chemically modified antibodies (LYTACs) or antibody-protein fusions (KineTACs). Prior to these well publicized efforts, there were several papers in 2016-2017 looking at bispecific ADCs (binding to HER2 plus TfR, CD63, or the prolactin receptor) attempting to enhance their efficacy by driving enhanced internalization. As was anticipated, these approaches also drove targeted degradation of HER2. This paper provides an additional example of a secondary binding event that can drive targeted degradation of a cell surface receptor.

The authors do suggest that these molecules are differentiated from bispecific Abs because they don't require assembly, and the LYTAC technology due to the lack of chemical conjugation. While technically true, many of these processes are very well established, and the KineTACs, in particular, could be rearranged to provide a similar outcome. More intriguing to me is that the presumed co-receptor that binds to the C-terminus of PCSK9 (PCTD) seems to be endo/lysosomal rather than on the cell surface. As a result, these molecules could, in principle, avoid some of the biodistribution risks seen for the other approaches. The caveat, of course, is that the detailed mechanism of internalization is not described and the presumed co-receptor is not identified.

Broadly speaking, the approach appears comparable to many of the other published approaches. The authors report partial degradation of a small number of targets, with the efficiency of degradation varying by target. The controls and experimental designs are reasonable but are predominantly conducted on engineered rather than native systems. The biology addressed in the paper changes from figure to figure providing limited biological understanding or impact.

A few suggestions for improving the manuscript...

- 1) Identification of a co-receptor(s) for the PCTD would greatly boost the impact of this publication both by adding some novel biological discovery of the natural PCSK9 pathway, and establishing the mechanism by which this approach works.
- 2) Additional characterization of the mechanism of degradation would help. What are the kinetics? Does it reach equilibrium? What are the determinants of effective degradation (antibody affinity, epitope, fusion structure, etc.?)
- 3) Greater focus on endogenous rather than transfected US28 would improve the relevance of the work, as would some more depth into what the precise impact of US28 degradation is in HCMV infection both in myeloid and fibroblast cells.
- 4) The use of the Kasumi-3 line is somewhat limiting. Crawford et al (mBio 2022) write "Importantly, dysregulated signaling due to the transformed nature of both cell types (Kasumi-3 and THP-1) presents hazards for studying the cell signaling pathways that regulate latency and reactivation." CD34+ iPSCs or perhaps primary monocytes could provide more convincing responses on non-immortalized cells.
- 5) Of minor importance, but the field has come up with endless acronyms for various degradation technologies. I would recommend not adding to the list. As is the "TAC" portion of PACTACs here is for "targeted clearance" rather than "targeting chimeras" as was initially proposed by Craig Crews. I would suggest changing the meaning of this portion of the acronym is inviting confusion.

Referee #1:

The paper by Schmitt and colleagues describes a mechanism for inducing the degradation of cell-surface proteins using modified antibodies. Antibodies are expressed with a PCSK9 domain that elicits target degradation via the endolysosomal system. The authors demonstrate convincing degradation of a few targets, including TfR and HCMV protein US28. The work appears sound and convincing. It is well written and easy to follow. The major caveat, about which the authors are candid, is that it is unknown how PCSK9 is directing target degradation. However, this is a new technology that could find applications in biomedical research without a full understanding of its mechanism.

[REDACTED: Referee's comment and the author's response with unpublished data.]

Figure 4 - the results suggest that PACTAC against US28 does not induce virus replication, only IE expression. Is this linked to the apparently low levels of degradation of US28? Would complete US28 abrogation cause lytic infection?

The referee poses a valid question. It is well established (by us and others) that infection of CD34+ progenitors (Humby and O'Connor, 2015) and CD14+ monocytes (Krishna et al 2017) with HCMV in which US28 has been deleted (HCMV Δ US28) not only results in IE gene expression but production of infectious virus. However, targeting US28, once latency has been established, using antagonistic antibody to US28, also only induced partial reactivation with induction of IE expression but no production of infectious virus. One interpretation of this could be that the presence of US28 at the earliest times of infection (as an incoming viral structural protein) establishes a cell phenotype which not only represses IE expression but also helps suppress Early (E) and Late (L) gene expression. In contrast, removing US28 expression, once latency has been established, derepresses IE expression but not E and L promoter activity. Alternatively, the referee could be correct and the extent of removal of US28 by degradation could be the key issue, here. We have addressed this by discussion in the revised manuscript.

Referee #2:

The authors report a new method for targeted degradation of cell-surface proteins that exploits the natural function of the endogenous protein PCSK9. In this method, cell surface proteins are targeted with a chimeric molecule consisting of target-binding antibody/antibody fragment fused to the C-terminal domain of PCSK9, which they call PACTACs. Although the mechanism of endogenous PCSK9 action is poorly understood, the authors show that their PACTAC molecules cause lysosomal re-routing of membrane proteins that usually recycle through endosomes back to the cell surface. The authors prove the effectiveness of PACTACs against several reporter proteins as well as the endogenous protein Transferrin. Finally, they design a PACTAC against the human cytomegalovirus encoded protein US28 and show that this approach has therapeutic potential for latent HCMV infection.

The key finding of this manuscript is the use of the C-terminal domain of PCSK9 as a degron that can be fused to antibodies or nanobodies to induce lysosomal degradation of cell surface proteins. This finding is significant as it represents a novel addition to the growing number of methods for targeted protein degradation, which will be of interest to a broad audience. Overall the manuscript is well-written and the findings are well-supported by high-quality experimental data. I recommend publication after the following minor comments are addressed:

Minor comments:

1) Figure 1e,f and Figure 4: It would benefit the reader if the authors include illustrations of the PACTACs used in these experiments. This could be structural models similar to what is already shown in Figure 1a,b and 2b or even simple cartoon illustrations. For example, from just the text it is hard to visualize what the OKT9FabPCTD construct might look like.

We have now included cartoon schematics in Figure 1 to illustrate the anti-GFP and OKT9Fab constructs.

2) Figure S3a: For the labels the authors should include + sign between US28-GFP and the PACTACs to illustrate that these conditions are HCMV-US28-GFP infected cells plus treatment with e.g. PCTD-Vun100bv. The text/numbers on the X and Y axis of these graphs is very small.

We have included all these suggested amendments in Figure EV4.

3) Figure 4: It would benefit non-virologist/immunologist readers if the authors include a schematic here to show how US28 represses IE2, and that PACTAC degradation of US28 would relieve this repression resulting in IE2-positive cells, which in turn triggers T-cell killing.

The referee raises a valid point. We have included a schematic (now Figure 5A) that illustrates the role of US28 in latency, how PCTD-Vun100bv induces IE gene expression, and how IE-expressing cells are recognized and killed by cytotoxic T-cells.

Referee #3:

Schmitt et al. describe the development of "PACTACs" (PCSK9-Antibody Chimeras for Targeted Clearance) which allow the targeted degradation of internalizing cell surface proteins. There has been substantial recent interest in this field, including descriptions of bispecific antibodies that drive targeted ubiquitination and subsequent degradation (AbTACs/PROTABs) and, similar to this manuscript, molecules that drive increased internalization and endolysosomal degradation via chemically modified antibodies (LYTACs) or antibody-protein fusions (KineTACs). Prior to these well publicized efforts, there were several papers in 2016-2017 looking at bispecific ADCs (binding to HER2 plus TfR, CD63, or the prolactin receptor) attempting to enhance their efficacy by driving enhanced internalization. As was anticipated, these approaches also drove targeted degradation of HER2. This paper provides an additional example of a secondary binding event that can drive targeted degradation of a cell surface receptor.

The authors do suggest that these molecules are differentiated from bispecific Abs because they don't require assembly, and the LYTAC technology due to the lack of chemical conjugation. While technically true, many of these processes are very well established, and the KineTACs, in particular, could be rearranged to provide a similar outcome. More intriguing to me is that the presumed co-receptor that binds to the C-terminus of PCSK9 (PCTD) seems to be endo/lysosomal rather than on the cell surface. As a result, these molecules could, in principle, avoid some of the biodistribution risks seen for the other approaches. The caveat, of course, is that the detailed mechanism of internalization is not described and the presumed co-receptor is not identified.

Broadly speaking, the approach appears comparable to many of the other published approaches. The authors report partial degradation of a small number of targets, with the efficiency of degradation varying by target. The controls and experimental designs are reasonable but are predominantly conducted on engineered rather than native systems. The biology addressed in the paper changes from figure to figure providing limited biological

understanding or impact.

A few suggestions for improving the manuscript...

1) Identification of a co-receptor(s) for the PCTD would greatly boost the impact of this publication both by adding some novel biological discovery of the natural PCSK9 pathway, and establishing the mechanism by which this approach works.

We agree that identification of a co-receptor for PCTD would be of considerable interest in terms of understanding both PCSK9 regulation of LDLR and the more general application described here. However, considerable effort has so far been expended by other laboratories in trying to identify such a co-receptor, and although several co-receptors have been proposed, there is currently no standout candidate. We therefore feel that definitive identification of the co-receptor is likely to require a considerable investment, and thus beyond the scope of the current paper.

2) Additional characterization of the mechanism of degradation would help. What are the kinetics? Does it reach equilibrium? What are the determinants of effective degradation (antibody affinity, epitope, fusion structure, etc.?)

We agree that further characterisation of the mechanism would be extremely valuable. To that end, we have investigated the kinetics of TfR degradation using the OKT9FabPCTD/OKT9Fab constructs in SK-BR-3 cells, a breast cancer cell line, to measure the rate of degradation and the equilibrium levels of TfR reached after prolonged treatment. These results have been incorporated into the revised manuscript, together with a dose-escalation experiment which provides further insight into the efficiency of degradation over a range of doses.

Regarding the potential determinants of effective degradation (antibody affinity, epitope and fusion structure), we note that a rigorous assessment of the first two factors (antibody affinity / epitope) would ideally require a panel of antibodies differing by affinity and/or epitope against the same target. Unfortunately, we are not currently in a position to generate such a panel. However, we note that the published affinities of the OKT9 fab (~4 nM, <https://www.ncbi.nlm.nih.gov/pmc/articles/PMC8354235/>) and the anti-BC2 nanobody (1.4nM, <https://www.nature.com/articles/srep19211>) are similar, whereas the degree of degradation between TfR and the BC2-tagged NPxY reporter construct differs (69% vs 34% respectively); thus, it seems likely that the trafficking characteristics of the target are a more significant determinant of efficacy than antibody affinity. We have added text to the paper to discuss this point. To address the impact of fusion structure on degradation, we cloned and expressed a variety of different OKT9FabPCTD fusion constructs, differing by fusion point and linker length, and tested their degradation efficiencies, and have included this new data in the revised manuscript.

3) Greater focus on endogenous rather than transfected US28 would improve the relevance of the work, as would some more depth into what the precise impact of US28 degradation is in HCMV infection both in myeloid and fibroblast cells.

The referee raises a valid point. We have performed additional experiments to investigate

whether US28 degradation upon PCTD-Vun100bv treatment induces IE gene expression in primary CD14+ monocytes, a site of HCMV latency in vivo. Pleasingly, PCTD-Vun100bv indeed induced ~2 times higher IE levels than the non-US28-targeting PCTD-αBC2 control or the background level observed in untreated cells (Figure 6B).

For fibroblasts, previous studies have shown that US28 is dispensable for efficient lytic viral growth in fibroblasts (Krishna et al, 2018). Therefore, we expect further studies on the impact of US28 degradation in fibroblasts to show little effect.

4) The use of the Kasumi-3 line is somewhat limiting. Crawford et al (mBio 2022) write "Importantly, dysregulated signaling due to the transformed nature of both cell types (Kasumi-3 and THP-1) presents hazards for studying the cell signaling pathways that regulate latency and reactivation." CD34+ iPSCs or perhaps primary monocytes could provide more convincing responses on non-immortalized cells.

We agree with the referee that the use of Kasumi-3 cells for HCMV latency and reactivation studies is somewhat limited – for example, treatment with HDAC inhibitors in Kasumi-3 cells and CD34+ hematopoietic stem cells caused different effects on IE1 transcription (Crawford et al, 2022). We addressed this in the revised manuscript and included data on PCTD-Vun100bv treatment in primary CD14+ monocytes that shows that PCTD-Vun100bv induces ~2 times higher IE expression levels than the non-US28-targeting PCTD-αBC2 control or the background level observed in untreated cells (Figure 6B).

5) Of minor importance, but the field has come up with endless acronyms for various degradation technologies. I would recommend not adding to the list. As is the "TAC" portion of PACTACs here is for "targeted clearance" rather than "targeting chimeras" as was initially proposed by Craig Crews. I would suggest changing the meaning of this portion of the acronym is inviting confusion.

We have changed the meaning of the acronym to 'PCSK9-Antibody Clearance-Targeting Chimeras' in line with this suggestion and for consistency with the PROTAC acronym.

Dear Dr. Kelly

Thank you for the submission of your revised manuscript to EMBO reports. It has been evaluated by former referee #3, who recommends publication.

Browsing through the manuscript myself, I noticed a few editorial things that we need before we can proceed with the official acceptance of your study.

- We will publish your manuscript in our Methods section. We would therefore ask you to use 'Structured Methods', our new Materials and Methods format, which is mandatory for Methods and Articles with a strong methodological focus. According to this format, the Materials and Methods section should include a Reagents and Tools Table (listing key reagents, experimental models, software and relevant equipment and including their sources and relevant identifiers) followed by a Methods and Protocols section in which we encourage the authors to describe their methods using a step-by-step protocol format with bullet points, to facilitate the adoption of the methodologies across labs.

More information on how to adhere to this format as well as downloadable templates (.doc or .xls) for the Reagents and Tools Table can be found in our author guidelines: <

<<https://www.embopress.org/doi/10.15252/msb.20178071>>.

- Please remove the Author Contributions from the manuscript file and make sure that the author contributions in our online submission system are correct and up-to-date. The information you specified in the system will be automatically retrieved and typeset into the article. You can enter additional information in the free text box provided, if you wish.

- Please add a callout to Fig. 4A-B in the text where appropriate.

- Source data files need to be saved as one folder per figure with subfolders for each panel. Each Figure folder is then uploaded as .zip file. E.g. all the Source data files for Figure 1 need to be saved in a single folder and this needs to be zipped and then uploaded as "SD figure 1.zip" file.

- The following funding information is listed in the manuscript but not in the online submission system: CIMR Flow Cytometry Core Facility; the Cambridge NIHR BRC Cell Phenotyping Hub. Please update this information.

- Please remove the ORCID identifiers from the front page of the manuscript text. It is sufficient (and necessary) to specify these in the online submission system. The ORCID IDs are currently missing for John Sinclair and David Owen.

- The Ethical Approval should be part of the Materials and Methods section. Please supply the reference number for approval.

- The Data availability paragraph comes after the Materials and Methods section. The correct order here is: Materials and Methods, Data availability, Acknowledgements, Disclosure and competing interests statement

- Please add a statement on mycoplasma testing and/or cell line authentication in the methods section (currently only stated in the Author Checklist).

- Figure 4, 5, 6,: in case the panels show data from one representative experiment, as the legend indicates, the statistical analysis should be removed, since $n = 1$, even though you pooled technical replicates. I recommend showing the means and SD across the biological replicates, instead (plus statistical analysis). As per our editorial policies statistical analysis may only be applied if the data are obtained from at least 3 biological (not technical) replicates (Fig 1D e.g., is derived from technical replicates).

- Please supply the synopsis image as .jpg or .png file at a final size of 550 pixels width. Please also consider if the two data panels at the bottom left and right are required or whether the schematic representation of the approach might be sufficient.

- Our production/data editors have asked you to clarify several points in the figure legends (see below). Please incorporate these changes in the manuscript and return the revised file with tracked changes with your final manuscript submission.

a) Please note that a separate 'Data Information' section is required in the legends of figure 5; 6. I.e., Add "Data information: " to the paragraph on "Statistical analysis were performed using the one-way.." and to "Statistical analyses were performed using...".

b) Please note that in figure 1d there is a mismatch between the annotated p values in the figure legend and the annotated p values in the figure file that should be corrected.

c) Please define the annotated p values ** in the legend of figure 1f as appropriate.

d) Please indicate the statistical test used for data analysis in the legend of figure 1f.

- On a different note, I would like to alert you that EMBO Press offers a new format for a video-synopsis of work published with us, which essentially is a short, author-generated film explaining the core findings in hand drawings, and, as we believe, can be very useful to increase visibility of the work. This has proven to offer a nice opportunity for exposure i.p. for the first author(s) of the study. Please see the following link for representative examples and their integration into the article web page:

<https://www.embopress.org/doi/full/10.15252/emj.2019103932>

With kind regards,

Referee #3:

The authors have adequately addressed the suggestions highlighted in the first review.

The authors addressed the minor editorial issues.

Dr. Bernard Kelly
University of Cambridge
Cambridge Institute for Medical Research
Hills Road
Cambridge, Cambridgeshire CB2 0XY
United Kingdom

Dear Bernard,

I am very pleased to accept your manuscript for publication in the next available issue of EMBO reports. Thank you for your contribution to our journal.

Kind regards,

Martina
